# Bromine Speciation in Volcanic Plumes: New in-situ Derivatization LC-MS Method for the Determination of Gaseous Hydrogen Bromide by Gas Diffusion Denuder Sampling

Alexandra Gutmann[1,2,3], Nicole Bobrowski[3,4], Marcello Liotta[5], Thorsten Hoffmann[1]

[1]Department Chemie, Johannes Gutenberg-University, Mainz, Germany
[2]Institut of Geosciences, Johannes Gutenberg-University, Mainz, Germany
[3]Max-Planck Institute for Chemistry, Mainz, Germany
[4]Institute for Environmental Physics, University of Heidelberg, Heidelberg, Germany
[5]Instituto Nazionale di Geofisica e Vulcanologia, Sezione di Palermo, Italy

*Correspondence to*: Alexandra Gutmann (alexandragutmann@web.de), Thorsten Hoffmann (t.hoffmann@uni-mainz.de

**Abstract.** The chemical characterization of volcanic gas emissions gives insights into the interior of volcanoes. Bromine species have been correlated with changes in the activity of a volcano. In order to exploit the volcanic bromine gases, we need to understand what happens to them after they are outgassed into the atmosphere.

This study aims to shed light on the conversion of bromospecies after degassing. The method presented here allows the specific analysis of gaseous hydrogen bromide (HBr) in volcanic environments. HBr is immobilized by reaction with 5,6-epoxy-5,6-dihydro-[1,10]-phenanthroline (EP), which acts as an inner coating inside of diffusion denuder tubes (*in situ* derivatization). The derivative is analyzed by high-performance liquid chromatography coupled to electrospray ionization mass spectrometry (HPLC-ESI-MS).

The collection efficiency for HBr (99.5 %), collection efficiency for HBr alongside HCl (98.1%) and the relative standard deviation of comparable samples (8 %) have been investigated. The comparison of the new denuder-based method and Raschig Tubes as alkaline trap resulted on average in a relative bias between both methods of $10 \pm 6$ %.

The denuder sampling setup was applied in the plume of Masaya (Nicaragua) in 2016. HBr concentrations in the range between 0.44 and 1.97 ppb were measured with limits of detection and quantification below 0.1 and 0.3 ppb respectively. The relative contribution of HBr as a fraction of total bromine decreased from $75 \pm 11$ % at Santiago rim (214 m distance to the volcanic emission source) to $36 \pm 8$ % on Nindiri rim (740 m distance).

A comparison between our data and the previously calculated HBr, based on the CAABA/MECCA box model, showed a slightly higher trend for the HBr fraction on average than expected from the model. Data gained from this new method can further refine model runs in the future.

## 1 Introduction

Since the formation of the atmosphere, volcanic eruptions and volcanic passive degassing contribute to an exchange of Earth's interior and the atmosphere (Brown, 1952; Schmincke, 1993; Halmer et al., 2002) with impacts extending from the local environment to the global climate (Shaw, 2008; Saturno et al., 2018). Besides their environmental and climatic impacts, volcanic gases can be indicators for processes occurring within volcanoes. For example, in 1991, volcanologists observed a significant increase of sulfur dioxide ($SO_2$) emissions at Mt. Pinatubo which influenced the decision to evacuate ten thousands of people prior to the onset of major volcanic activity (Daag, 1996; Harlow et al., 1996).

Besides $H_2O$, $CO_2$ and sulfur species, also halogens are constituents of volcanic emissions e.g. in arc volcanoes 0.84% HCl, 0.061% HF, 0.0025% HBr (Textor et al., 2004; Gerlach, 2004). Monitoring of halogens in volcanic plumes has been correlated with volcanic activity for decades. For example, Noguchi and Kamiya determined the plume composition of Mt. Asama over months and observed a decreasing Cl/S ratio before an eruption in 1958 (Noguchi and Kamiya, 1963). The discovery of bromine monoxide (BrO) in volcanic plumes and the observed correlation between the simultaneously determined $BrO/SO_2$ ratio and volcanic activity by automated instruments (e.g. Lübcke et al., 2014) made $BrO/SO_2$ a promising monitoring tool. A major advantage of BrO is a typically negligible background in the atmosphere, concluding that findings clearly derive from magma degassing. The $BrO/SO_2$ ratio is measurable simultaneously by remote sensing instruments which facilitates the applicability for monitoring volcanic activity compared to in situ sampling techniques. Even inaccessible volcanoes can be observed with satellite-based remote sensing instruments (e.g. Hörmann et al., 2013). If the informative value of Br emissions about volcanic activity is confirmed, then it may be possible to provide greater forewarning regarding volcanic eruptions via BrO monitoring through the use of remote sensing instruments.

It is therefore important to have methods to determine how bromine species are transformed in volcanic plumes. Gas composition and gas amount of distinct halogen species vary not only due to volcanic activity changes but also because of photochemical and multiphase reactions in the plume. It is assumed that bromine is mainly released from volcanoes as hydrogen bromide (HBr) and, depending on various conditions (e.g. total bromine emission, plume-air-mixing), is gradually converted into other bromine species (including BrO) in the volcanic plume. To use the $BrO/SO_2$ ratios for monitoring volcanic activity, a perfect understanding of ongoing bromine chemistry in the plume is essential. Chemical models have already been developed to explain BrO formation in certain plume environments. However, due to the lack of applicable measurement techniques for individual bromine species, comparison of expected bromine species conversion rates with measurements are still hypothetical (Gutmann et al., 2018 and references therein). A promising approach for the revelation of the bromine speciation are gas diffusion denuder techniques. The gas diffusion denuder technique has already been developed for reactive halogens (e.g. $Br_2$, BrCl, HOBr) (Huang and Hoffmann, 2008, 2009) and has been adapted to volcanic plumes (Rüdiger et al., 2017) and even applied using UAV in volcanic plumes (Rüdiger et al., 2018). The gas diffusion denuder technique differentiates between gaseous compounds and compounds in the particle phase, e.g. HBr and particulate bound bromides (Kloskowski et al., 2002). To function as a sampling unit, the denuder surface should be a perfect sink for the compounds to

be analyzed (i.e. assuming an irreversible uptake), which diffuses to the coated inner walls of the denuder as the sampling air passes through the device. The sampling efficiency depends on the denuder dimensions, the flow velocity and the diffusion coefficient of the analyte of interest, but is > 99 % under normal sampling conditions (Fick, 1855; Gormely, P.G. and Kennedy, M., 1948; Townsend, 1900; Tang et al., 2014). Which gaseous analytes are enriched, depends on the organic reagent used as a coating for the denuder tubes. Epoxides are effective reagents for a rapid reaction with gaseous HBr, since they show acid-catalyzed ring-opening reactions with nucleophilic reagents, in the case of HBr both properties being present in one molecule. Therefore, as products from the reaction of epoxides with HBr bromohydrins are formed (Becker and Beckert, 2004).

Since the reaction products (besides bromohydrins also other halohydrins) are formed during sampling and are finally separated by liquid chromatography after extraction from the denuder tubes and are detected and quantified by mass spectrometry, the presented technique represents an *in situ* derivatization high-performance liquid chromatography-electrospray ionization mass spectrometry (HPLC-ESI-MS) method.

In this study, the aim was to develop a measurement technique with which it is possible to detect gaseous HBr in volcanic gases. Enrichment of gaseous HBr by chemical bonding to a coating in denuders seemed the most promising method.  First, a source of HBr test gas was developed, which is included in the methods section. Followed by the experimental setup, handling of the denuders, analysis of the samples, and finally the description of the application of the method in Masaya's plume in 2016. After testing five different coatings, 5,6-epoxy-5,6-dihydro-[1,10]-phenanthroline (EP) showed the best conditions for a successful method development. The performance of the EP-coated denuders is described in more detail in the following. In order to see to what extent a supplementation of simultaneously collected field samples with other methods is possible, a comparative measurement of the new denuder method was compared with alkaline samples from a Raschig tube. Finally, the results of the new method applied in the volcanic plume of Masaya volcano are presented.

## 2 Methods

For the development of a method based on the reaction with an epoxy coating, the collection efficiency of five different coatings was tested. All reagents used are listed in the supplementary material. The substances 2-bromocyclooctanol, 2,3-epoxy-3-phenylpropanoic acid, 9,10-epoxystearic acid, 10-bromo-9-hydroxystearic acid, and 5-bromo-6-hydroxy-5,6-dihydro-[1,10]-phenanthroline (EPBr) were required as coatings or reference products, were not commercially available, and were synthesized. Synthesis descriptions are also included in the supplementary material.

### 2.1 Test gas sources and experimental setup

#### 2.1.1    Test gas sources

A diffusion gas source using a brown glass vial with septum cap as described in Rüdiger et al. (2017) was filled with 48 % aqueous HBr and stored under nitrogen flow at 30 or 50° C (Fig. 1a). The incoming gas stream was thermostated before it reached the diffusion source. The vials containing the analytes were weighed regularly to determine output rates. The length

and diameter of the capillary, as well as the temperature, controlled the output rate. A 5 cm capillary with 0.32 mm inner diameter was used for the HBr source. Taking advantage of the azeotropic behaviour of the HBr-water-mixture (Haase et al., 1963) we observed a constant output rate for HBr of 108 µg/d and 415 µg/d for 30 and 50 °C respectively (Fig. S2).

For experiments testing the collection efficiency of HBr aside hydrogen chloride (HCl), a vial following the same procedure using a 2 cm capillary with 0.64 mm inner diameter was prepared containing 30 % aqueous HCl. We observed an output rate for HCl of 4.33 mg/day at 50 °C. For the experiments both vials (one each for HBr and HCl) were stored side by side in the test gas storage vessel (Fig 1a).

For $Br_2$, the permeation source described in Rüdiger et al. (2017) was used with an output rate of 775 µg/d under nitrogen flow.

### 2.1.2   Experimental setup

For a set of experiments, the test gas sampling was performed with two (or three) denuders in a row using a membrane pump (Gilian Gil Air Plus) located downstream of the denuder (Fig. 1a). A sampling flow rate of 250 mL/min was used. All connecting pieces were made of glass or PTFE material.

A method often used in volcanic emission studies is sampling with alkaline traps. We used so-called Raschig Tubes (Wittmer et al., 2014). These are horizontally aligned, rotating glass vessels containing Raschig rings, which are filled with an alkaline solution through which the sample air is sucked by means of membrane pumps. This technique was used here both as a comparative method in the laboratory to compare with the denuder technique (Fig, 1b) and later in the field for total bromine determination. The target HBr concentrations were generated by a standard gas bottle with $102.8 \pm 3.1$ ppm HBr diluted with 3-neck flasks. Care was taken to ensure that the connecting tubes to the entrance of the two sampling arrangements (Raschig Tube and denuder) were identical in length and diameter. This ensures that results of both sampling methods are comparable as concentration variations affect both methods equally. Although HBr concentrations between 13 and 31 ppb were established in this way, the downstream analysis revealed concentrations between 3 and 20 ppb. The cause is probably the loss of HBr on the surfaces of the experimental setup (e.g. Hanson and Ravishankara (1992), Talukdar et al. (1992), this applies generally to strong acids e.g. $HNO_3$ (Neuman et al., 1999)). Another inaccuracy of HBr concentrations may also come from incorrect flow meters or fluctuating gas flows.

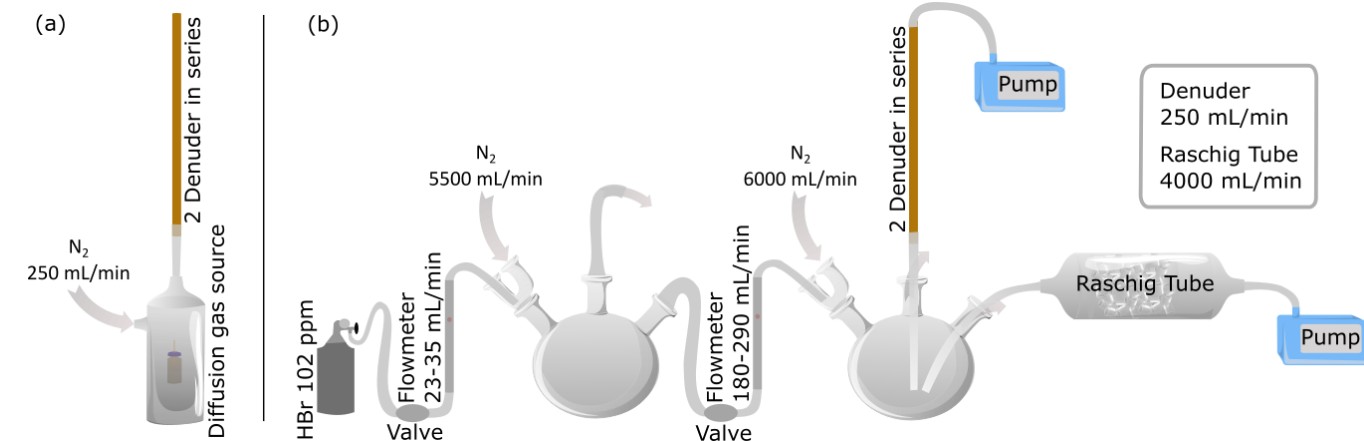

**Figure 1: Experimental sampling setups in the laboratory. (a) Denuders were connected to a diffusion test gas source with PTFE tubes. The test gas source was heated to 50° C. Entering nitrogen was thermostated before it reached the diffusion source. (b) Experimental Setup for comparison of denuder and Raschig Tube technique for HBr determination.**

## 2.2 Denuder preparation, extraction and analysis

The denuder were 50 cm long brown borosilicate glass tubes with an inner diameter of 6 mm. Solutions of 1.5 mmol/L, 7.5 mmol/L or 15 mmol/L were used with various coating compounds (Table 1). Besides the amount of coating material, the uniformity of the distribution of the coating inside the denuder influences the collection efficiency of each denuder. To achieve an even coating the coating solution was applied in 6 steps of 0.5 ml. Each aliquot was dried before the application of the next step. In order to obtain reproducible coating distributions and to standardize the coating process, a system was installed which can hold four denuders horizontally at an adjustable angle of about 10°, to connect the denuders with a gentle stream of nitrogen of about 0.5 L/min/denuder and to rotate them during preparation. A photo of the created drying system is shown in Fig. S3. The coated denuders were closed with polypropylene caps and sealed with PTFE tape and parafilm.

Coating agents for derivatization must have the following properties: The reaction between the coating and gaseous analyte must be sufficiently rapid to achieve high collection efficiencies. Furthermore, the reaction should not require the use of solvents or other additional chemicals. Fixation of coating reagents to the denuder walls during sampling can be done with glycerine (Finn et al., 2001) and was tested for 1,2-epoxycyclooctane-coated denuders. The coating should preferably provide only one derivatized product (no isomers or multiple derivatizations). In addition, the coating substrate must be suitable for the coating process. Carboxylic acids were used because they entail low volatility and thus do not evaporate during sampling and the concentration step. In the further course, attention was paid to low volatile compounds containing a functional group that exerts a positive inductive effect on the epoxy group to maintain reactivity to HBr were selected to optimize the reactivity of the coating.

After sampling, analytes were dissolved from the denuders in five steps using 2 mL solvent each step (Fig. 2a). All analytes were dissolved with ethyl acetate, except for EP-coated denuders, as these showed better solubility in methanol. The extraction

efficiency was investigated with EP-coated denuders doped with 0.01 µmol of the bromine product EPBr. Therefore, standard calibration solution has been applied and dried on the denuder during the denuder coating process. The extraction process described in Fig. 2a has been performed a second time on a previously extracted denuder. Analyzing the residue of EPBr in the second extraction step, less than 0.05 % EPBr compared to the first step was found.

Extracts were concentrated to approximately 100 µL under a gentle nitrogen stream at 35° C. For adjustment of varying
evaporated volumes and for compensation of evaporation losses, 100 µL of an internal standard was added to the extracts before the concentration step. Samples analyzed by GC-MS were doped with 100 µL 2,4,6-tribromanisole (TBA, 6 mg/L) as internal standard, while EP-coated denuder samples analyzed by LC-MS were doped with 100 µL neocuproine (NC, 5 mg/L). The suitability of the internal standards was evaluated by investigating the recovery rate.

The recovery rate of the processing method for EPBr showed recovery rates ranging from 110 to 51%. We observed that the
155 high amounts of EP cause precipitation when samples are concentrated for analysis. The addition of formic acid enhances solubility of analytes while not affecting negatively the following LC-ESI-MS analysis. The recovery for EPBr was determined to be $99 \pm 4$ % when formic acid was added to the extracts before the evaporation process (Fig. 2b).

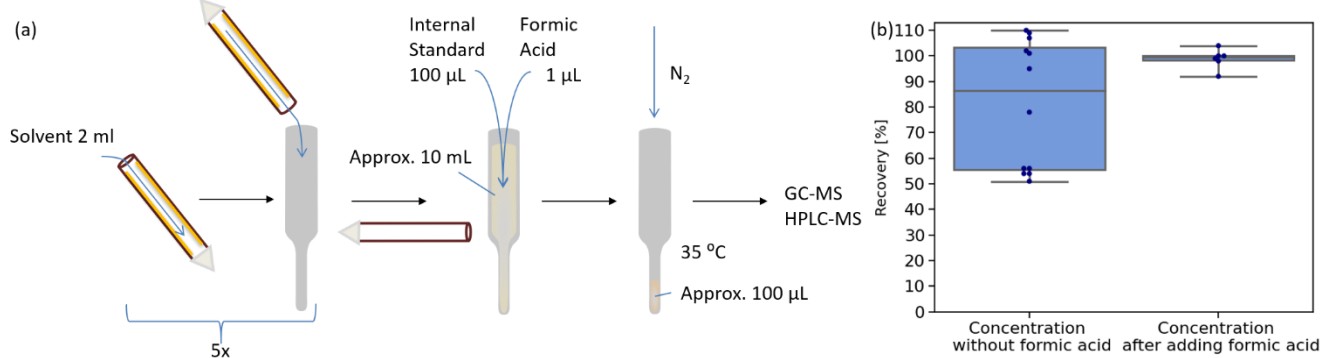

**Figure 2: Denuder coating extraction and concentration of samples. (a) Denuder coating is dissolved 5 times with 2 ml solvent each**
160 **step. Dissolved coating is collected in glass vessels. 100 µl internal standard (NC for EP-coated denuders, TBA for all others) and 1 µl formic acid for EP-coated denuder samples is added to the approximately 10 ml coating solution before concentration. The samples are concentrated at 35 °C under a gentle nitrogen stream to approximately 100 µl. (b) The investigation of the concentration process without adding formic acid (left side) revealed a mean recovery for EPBr of $81 \pm 25$ % and a median (grey line) at 86%. Addition of formic acid before the concentration (right side) enhances recovery of EPBr to $99 \pm 4$ %. The boxes show the 25th and**
165 **75th percentiles and the median.2.3 Analysis by chromatography-mass spectrometry**

All samples were analyzed by GC-MS, except for EP-related samples, which were analyzed by LC-MS. For all bromine compounds, both isotope masses were measured and taken into account for data processing.

### 2.3.1 Gas chromatography-mass spectrometry

To increase the volatility of carboxylic acids, the carboxylate functionality was converted to a trimethylsilyl derivative by
170 adding 30 µL *N,O*-bis(trimethylsilyl)trifluoroacetamide (BSTFA) and 7 µL pyridine to GC-samples containing carboxylic acids. Samples were incubated for 90 min at 70° C before analysis.

For chromatographic analysis, a 6850 Network GC System (Agilent, Waldbronn, Germany) with a fused-silica FS-Supreme-5ms capillary column (30 m x 0.25 mm i.d.; 0.25 μm, CS-Chromatography, Langerwehe, Germany) was used, coupled to a 5973 Network Mass Selective Detector (Agilent, Waldbronn, Germany). The carrier gas was 99.999 % high-purity helium at a constant pressure of 100kPa. 1 μL sample was injected in splitless mode. The injector was heated to 250° C and the transfer line to 300 °C. The Electron Ionization (EI) spectra were acquired at 70 eV and a mass range of *m/z* 45-450 were selected. The oven temperature programs (range from 90-300 °C with total runtimes of 9-30 min) and the ions used for quantification are summarized in the supplementary material (Table S1).

### 2.3.2 High-performance liquid chromatography-mass spectrometry

Since EP decomposes at usual GC-temperatures, EP-related samples were analyzed by high-performance liquid chromatography coupled to electrospray ionization mass spectrometry (HPLC-ESI-MS). HPLC-ESI-MS was performed with an Agilent 1100 series (Agilent Technologies, Germany) HPLC system coupled to an HCT-Plus ion trap mass spectrometer (Bruker-Daltonics, Germany). The analytical column (Atlantis T3 C18 2.1x150 mm, 3 μm particle size, Waters, Germany) was heated to 35° C during analysis. A flow of 200 μL/min was used. A gradient of eluent A (ultrapure water with 2 % acetonitrile and 0.04 % formic acid) to eluent B (acetonitrile with 2 % ultrapure water) was used, starting with 5 % eluent B that was held for 10 min. Within 23 min eluent B was raised to 14 % and in the following minute raised to 100 % (hold for 11 min). The electrospray ionization source was used in the positive ion mode and resulted in protonated molecular ions ($[M+H]^+$) which were used for quantification. The capillary voltage was +3.5 kV, the dry gas temperature was 350° C, dry gas flow ($N_2$) was 8.5 L/min, and nebulizer pressure 35 psi. The mass spectrometer was operated in ultra-scan mode.

Under these conditions, EP eluted at 7.9 min retention time, however, with relatively broad peak widths due to the high concentration of the coating material in the concentrated samples. 5-chloro-6-hydroxy-5,6-dihydro-[1,10]-phenanthroline eluted at retention time 20.5 min. In the field samples, the peak widths increased and retention times changed, probably due to overloading of the column. EPBr eluted at retention time 27.2 min. Depending on the amount of coating material and chloro-derivative, the retention time varied between 25 and 28 min. The internal standard neocuproine eluted at retention time 32.4 min (Fig. S5 and S6). A six-point calibration (0.1-10 mg/L) was used to determine the detection limit (LOD) of 0.03 mg/L and the quantitation limit (LOQ) of 0.17 mg/L for EPBr following a standard protocol (DIN 32645:2008-11.).

### 2.4 Field application at Masaya 2016

A first set of field samples was collected between 18.-21. of July 2016 at the Santiago Crater of the Masaya volcano (Nicaragua). A detailed description of the location can be found in Rüdiger et al. (2021). In summary, sets from different methods were collected simultaneously together at changing locations with various distances (200-2000 m, Fig. 3b) to Masaya's emission source at Santiago crater (Fig. 3b).

A total of eight ground-based and two UAV-based samples for the newly developed denuder method are presented here. In ground-based sampling sets, two denuders were sampled in series (Fig. 3a). Both denuders were extracted and results were summarized. Sampling was performed by a Gilian GilAir Plus handheld pump (battery included) with a flow rate of 250 ml/min for about 1-1.5 hours for each denuder. In addition to EP-coated denuders, samples with 1,3,5-trimethoxybenzene coated denuders for the determination of reactive bromine. as well as Raschig Tubes as alkaline traps for the determination of total bromine and total sulfur were collected simultaneously side by side. The results of these samples can also be found in Rüdiger et al. (2021).

First drone-based samples were collected with an UAV using a small four-rotor multicopter with foldable arms (Black Snapper, Globe Flight, Germany) called RAVEN (Rüdiger et al., 2018). For the UAV-based sampling, a remotely controlled sampler (called Black Box) was used and is also described in detail in Rüdiger et al. (2018). The Black Box enabled logging of the sampling duration and $SO_2$ mixing ratios via the built-in $SO_2$ electrochemical sensor (CiTiceL 3MST/F, City Technology, Portsmouth, United Kingdom). The Black Box has 20x14x13 cm. With this setup (Black Box + denuder) of approx. 1 kg we achieved flight times of up to 15 min. In drone-based sampling flights, individual denuders were used with sampling times between 5-10 minutes.

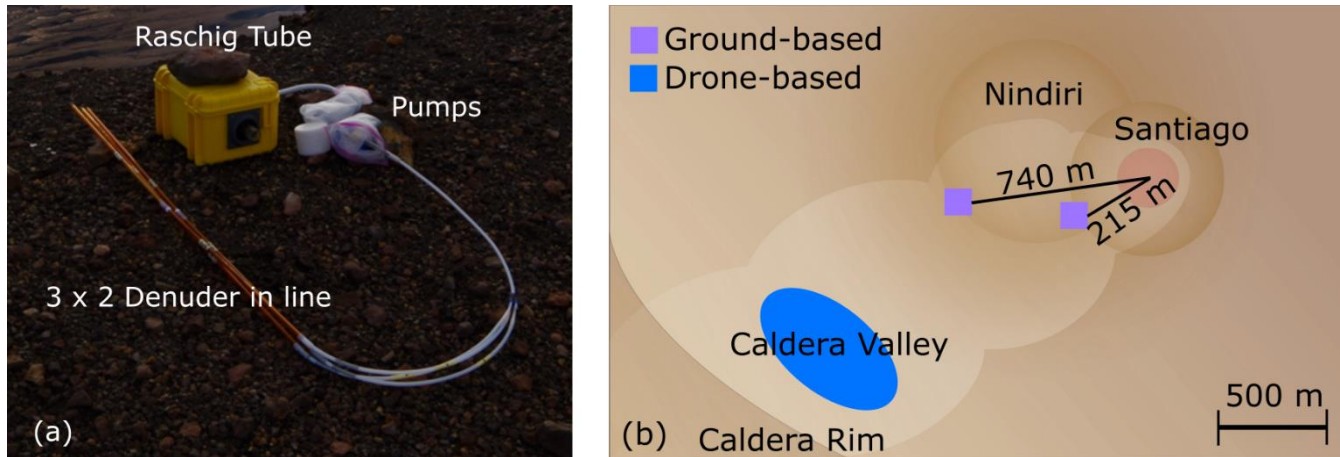

Figure 3: Field campaign at Masaya volcano in July 2016. (a) Sampling setup containing TMB- and EP-denuders, two in series each, and Raschig Tube. (b) Overview on ground-based sampling locations (purple areas) during the field campaign at Santiago crater, on Nindiri rim and UAV-based (blue area) in the caldera valley.

## 3 Results and Discussion

### 3.1 Selection of derivatizing coating

For the development of a method based on the reaction with an epoxy coating, the collection efficiency of five different coatings was tested. The observations of the different denuder coatings are summarized in Table 1.

Although it is solid at room temperature, 1,2-epoxycyclooctane was not detectable in samples extracted from denuders after sampling. Nevertheless, traces of the product 2-bromocyclooctanol could be detected in the denuder extracts. We did not quantify the formation, but this led us to believe in the principal ability of epoxides to react with HBr in our system.

The introduction of carboxy groups into the molecular framework of organic compounds reduces the vapor pressure of the substances. Therefore, epoxides with carboxyl groups were subsequently tested as coating materials. The selection of
230 compounds with carboxyl groups in the immediate vicinity of the epoxy functionality reduces the reactivity to HBr and accordingly no bromine product of *trans*-oxirane-2,3-dicarboxylic acid and 3-phenyloxirane-2-carboxylic acid could be detected.

The carboxyl group in 9,10-epoxystearic acid also reduces volatility but is not near the epoxy group. Technically, two isomeric products were possible, but the influence of the carboxylic acid group resulted in only one detected product (10-bromo-9-
235 hydroxystearic acid). When 9,10-epoxystearic acid-coated denuders were applied in the plume of Etna, a diol as a result of the reaction of the epoxide with water (9,10-dihydroxystearic acid) and the chloride derivative (10-chloro-9-hydroxystearic acid) were observed. Although derivatized HBr could be analyzed, the *m/z* ratios were overlaid by the water and chloride derivatives (same main *m/z* ratios, Fig. S4). The difficulties this posed for analysis led us to prefer EP, as it also reacted successfully with HBr in our system.
EP as a coating agent could retain and derivatize gaseous HBr passed through denuders and the bromohydrin product could be detected. The details of the characterization for this coating-compound are given in the following sections.

**Table 1: Selected epoxides used as coating reagentstable**

| Coating | Derivative | Observations |
|---|---|---|
| 1,2-epoxycyclooctane | 2-bromocyclooctanol | Evaporated during sampling |
| *trans*-oxirane-2,3-dicarboxylic acid | 2-bromo-3-hydroxy-butanedioic acid | No product |
| 2,3-epoxy-3-phenylpropanoic acid | 2-bromo-3-hydroxy-3-phenylpropanoic acid | No product |

| | | |
|---|---|---|
|  |  | Water and chloride side product: |
| 9,10-epoxystearic acid | 10-bromo-9-hydroxystearic acid | 9,10-dihydroxystearic acid |
|  |  | 10-chloro-9-hydroxystearic acid |
| 5,6-epoxy-5,6-dihydro-[1,10]-phenanthroline (EP) | 5-bromo-6-hydroxy-5,6-dihydro-[1,10]-phenanthroline (EPBr) | Chloride side product: |
|  |  | 5-chloro-6-hydroxy-5,6-dihydro-[1,10]-phenanthroline |
| | | Details are described in section 3.2. |

## 3.2 Denuder performance

### 3.2.1 Collection efficiency

According to Rüdiger et al. (2021) and Wittmer et al. (2014), about 0.5-5.9 ppb and 9.5-36 ppb total bromine was detected in ground-based samples in the volcanic plume of Masaya and Etna volcano respectively using alkaline traps. Besides HBr they detected HCl concentrations of 0.5-4.5 ppm and 0.1-20.6 ppm respectively. Other halogen species such as HCl react with EP via the same reaction pathways as HBr. But these form different derivatization products with EP and can, therefore, be easily distinguished by mass spectrometry. Still, HCl can consume the coating reagent. Estimating the speed of the derivatization reaction the nucleophilic reactivity of different hydrogen halides shows that bromide has higher nucleophilic reactivity than chloride (nucleophilic constants in water based on glycidol, $H_2O$: 0.00, $Cl^-$: 3.04, $Br^-$: 3.89, $I^-$: 5.04) (Swain and Scott, 1953). Accordingly, one could expect a higher reactivity of HBr compared to HCl.

Ensuring that the method will be able to retain all potential gaseous HBr even in high concentrated plumes the breakthrough behavior for 0.2 µmol HBr was investigated (0.2 µmol HBr correspond to 1 ppm HBr for 1 h sampling duration). Further, the collection efficiency for HBr was tested in the presence of about 5 µmol HCl (5 µmol HCl correspond to 4 ppm HCl for 1 h sampling duration).

The collection efficiency was tested with two or three denuders connected in series (Table 2). The amount of product found on the second and third denuder was compared with the values of the first denuder. The collection efficiency tested for denuders coated with 7.5 mmol/L EP coating solution revealed a breakthrough of HBr since about 30 % of the amount of the first denuder was observed in the third denuder. In contrast, for denuders coated with 15 mmol/L EP coating solution, the breakthrough for 1 ppm HBr was below 1 %. In competition with HCl $1.9 \pm 0.4$ % of the bromine product was found in the second denuders.

Coating amounts above 45 µmol EP caused precipitation in concentrated samples during sample preparation (see section 2.2). Concluding from this, 15 mmol/L EP coating solutions were used to coat denuders a theoretical amount of 45.0 µmol EP. The second denuder in series during sampling ensures that we will at least notice a relevant breakthrough of analytes.

**Table 2: Experimental details for the determination of collection efficiency for denuders coated with 7.5 and 15.0 mmol/L EP coating**

| Experiment (Fig. 1a) | EP-Coating Solution [mmol/L] | EP theoretical on denuder [µmol] | EP concentrated to 100 µL [mmol/L] | Approx. applied gases: HBr [µmol] | HCl [µmol] | Breakthrough 2. denuder [%] | 3. denuder [%] |
|---|---|---|---|---|---|---|---|
| 3 denuder in series | 7.5 | 22.5 | 225 | 0.2 | None | $105 \pm 5$ | $30 \pm 5$ |
| 3 denuder in series | 15.0 | 45.0 | 450 | 0.2 | None | $0.6 \pm 0.4$ | <0.1 |
| 2 denuder in series | 15.0 | 45.0 | 450 | 0.2 | 5 | $1.9 \pm 0.4$ | - |

### 3.2.2 Matrix effects, precision, LOD and LOQ

The influence of abundant EP on the EPBr determination was investigated by a test series with 450 mmol/L EP (corresponds to EP concentrations in concentrated denuder samples) and EPBr concentrations in the range of 19 to 263 nmol/ml (n = 6, Fig. S7). We determined a relative bias between the both sample types of $2 \pm 3$ %. We concluded that due to the detection of individual mass fragments by the mass spectrometer and internal standard adjustment, the ability of the method to quantify the reaction product is maintained even with EP-matrix.

Repeated measurements of the same gas composition using the setup shown in Fig. 1b resulted in a relative standard deviation of 8 % (see Fig. S8)

To ensure that results remain from detected analytes but not a higher noise (Fig. S5), the LOD and LOQ for denuder samples have been determined from blank denuders. The LOD and LOQ were determined by 3 and 10-fold deviation (Kromidas et al., 1995) from coated denuders transported and stored in the same way as denuder samples (3 coated but not sampled denuders for the field samples presented here). A LOD of 0.1 mg/L and a LOQ of 0.3 mg/L were calculated for EPBr. Since LOD and

LOQ for HBr in the atmosphere depend on sampling time and sample volume after evaporation to concentrated samples, their values were calculated separately for each sample (Table 4).

### 3.2.3 Stability of extracted samples

We tested the stability of extracted and concentrated samples with field sample like approaches that were stored in the freezer at -4° C (Table S3 and Fig. S9). Within the first two months of storage no systematic loss could be observed when comparing the analyzed EPBr/NC ratios. After long term storage of 2-3 years remeasuring the field samples (listed in section 3.5) revealed an average loss of $0.03 \pm 0.01$ % EPBr per day (about 11 % loss after 1 year of storage).

### 3.2.4 Interferences

If the denuder coating reacts with other types of bromine, resulting in the same product, this would lead to HBr overdetermination. Elemental bromine ($Br_2$) is the most likely cross-interference. 9,10-epoxystearic acid and EP-coated denuder collected the output of the $Br_2$ source for one hour following the setup in Fig. 1a. No bromine product was found. Other bromine species such as bromine oxides (e.g. BrO) with their positively polarized bromine atoms, no nucleophilic attack on the epoxide reaction center leading to bromohydrin is expected.

**3.4 Comparison between denuder and Raschig method**

Alkaline traps determine the total bromine content, gaseous HBr is thus measured here as a part of the total bromine. Speciation of the individual bromine species is not possible with alkaline traps, but if only gaseous HBr is sampled by both methods in the laboratory, denuder method and alkaline traps should produce the same results. Therefore, a comparative measurement of the denuder method with a Raschig Tube as an alkaline trap was set up to check the accuracy of the newly developed denuder

method. The experimental setup is shown in Fig. 1b. In five experimental series, HBr concentrations between 3 and 20 ppb were determined simultaneously (Table 3). A Dean-Dixon outlier test was applied in order to evaluate possible outliers (Dixon, 1950). No outlier was identified for $\alpha=0.05$ (If the significance level was changed to $\alpha=0.01$, experiment 4 could be considered an outlier.). Consequently, all the results were taken into account.

On average, the results related to denuder sampling yielded $99 \pm 11$ % of the HBr values determined by the Raschig Tube. An

305 orthogonal distance regression was performed and is shown in Fig. 4. Based on the line equation obtained, small values, such as those observed here in the field samples, can yield higher results from denuder determinations than expected via the Raschig Tube. We have concluded, that the HBr values determined by denuders in field samples can be considered a fraction of the total bromine determined by the Raschig Tubes. To account for the comparison studied, the deviation found is included as an error of the denuder field samples in Table 4.

**Table 3: Comparison of simultaneously test gas sampling of EP coated denuders and Raschig Tubes for gaseous HBr determination. The setup is shown in Fig. 1b.**

| Experiment | Denuder | | Raschig | |
| --- | --- | --- | --- | --- |
| | Sampled amount of HBr [nmol] | Calculated HBr in sampled test gas [ppb] | Sampled amount of HBr [nmol] | Calculated HBr in sampled test gas [ppb] |
| 1 | 1.8 ± 0.6 | 3.6 ± 1.6 | 26.0 ± 3.6 | 4.1 ± 0.6 |
| 2 | 2.3 ± 0.7 | 5.8 ± 2.2 | 28.1 ± 3.9 | 5.1 ± 0.7 |
| 3 | 3.0 ± 0.4 | 7.7 ± 1.2 | 41.6 ± 5.3 | 7.4 ± 1.0 |
| 4 | 4.8 ± 0.7 | 14.4 ± 2.3 | 83.0 ± 11.7 | 16.8 ± 2.4 |
| 5 | 7.6 ± 0.7 | 18.5 ± 1.8 | 111.7 ± 14.7 | 17.9 ± 2.4 |

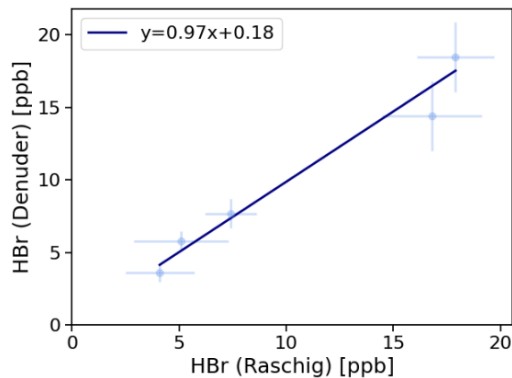

**Figure 4: Denuder results are plotted against Raschig results. The orthogonal distance regression model resulted in: y = 0.97 (± 0.10) * x + 0.18 (± 0.10) and a residual variance of 0.25.**

### 3.5 Field application at Masaya 2016

Here we present a first set of field samples using the new denuder method. The eight sets of ground-based measurements range
from 0.44 to 1.97 ppb (Table 4). Two UAV-based measurements are below their LOD. In UAV-based samples, the higher
LOD in UAV-based measurements result from a much shorter sampling time of 5-10 minutes (limited by maximum possible
flight time) compared to ground-based measurements (1-1.5 h).

To get an idea about the reproducibility of these field measurements a duplicate set of two EP-coated denuders each was
collected in parallel side by side on July 21. The parallel denuder measurements resulted in HBr concentrations of 1.97 ± 0.11
and 1.82 ± 0.10 ppb at the Santiago rim and 1.17 ± 0.07 and 0.97 ± 0.09 ppb at the Nindiri rim. The mean values and standard
deviations of 1.90 ± 0.11 and 1.07 ± 0.14 ppb of the two parallel samples result in a relative standard deviation of 6 and 13 %,
respectively. While the deviation from the Santiago rim samples is within those also found for laboratory samples (8 %),

possible causes leading to larger errors and affecting simultaneously collected samples differently may be passive diffusion during installation of the tubes or ash blowing in.

Based on the changes in the HBr concentrations at different distance from the emission source, we can observe a decrease in HBr concentrations with increasing distance. A cause for the decreasing concentrations can be, of course, a dilution of the plume. To compensate for this effect in our observations, we look at the ratio of the HBr to the total bromine detected by the Raschig Tube. Since both methods collected their samples side by side at the same time, we assume that a dilution of the plume effects equally both samples. The relevant Raschig Tube results for the total bromine from Rüdiger et al. (2021) are

summarized in Table 4. We have also adopted the estimated wind speed of 5 m/s.

Thus, we observed HBr fractions of 60 to 89 % of total bromine at Santiago Rim and 30 to 46% at Nindiri Rim. This results in an average of 75 ± 11 % HBr of the total bromine at a plume age of 0.7 min (214 m at the Santiago crater) decreases on average to 36 ± 8 % at 2.5 min (740 m at the Nindiri rim) further downwind (Fig. 5).

In the work of Rüdiger et al. (2021) the results of total bromine, total reactive bromine and bromine monoxide from

340 accompanying methods were used to run the atmospheric box model CAABA/MECCA, which was initialized by a high-temperature equilibrium model. The model run that best described the data in Rüdiger et al. (2021) was used here for comparison and is highlighted in light blue for the ratio of HBr/total Br in Fig. 5. This run was based on a Br/S ratio of 7.4 $\times 10^{-4}$. The Br/S ratio for the measurements considered here on 18.-21.7. was on averaged 6.2 ± 1.0 $\times 10^{-4}$. Even though the general trend between measured values and model predictions is consistent, on average the measured values appear to be

slightly higher than those calculated by this model run. Following the observations in Rüdiger et al. (2021), a cause may be the influence of aerosol. Aerosol was not measured simultaneously, smaller particle number concentrations and diameters than assumed may lead to slower HBr loss than expected. Also, deviation from the assumed wind speed can lead to a horizontal shift of the measurements while the deviations between Denuder and Raschig method cause a vertical shift. Overall, the trend observed would have to be confirmed by further samples. These samples give us a first idea that we can confirm our general

idea about the HBr consumption. Of course, a solid foundation will require many field samples and further consideration of the two methods used and their joint application at the expected concentrations.

**Table 4: Results of denuder measurements sampled in Masaya's plume on three days in July 2016. Sampling has been performed at three different locations with the following distances to the emission source: Santiago Rim 215 ± 50 m, Nindiri Rim 740 ± 50 m and**
355 **in the Caldera Valley 2000 ± 150 m (Fig. 3b). Total Bromine has been determined by simultaneously applied Raschig Tubes (details in Rüdiger et al. 2021). HBr concentrations (in ppb) were determined by EP-coated denuders. Their respective LOD and LOQ were calculated based on the signal-to-noise approach using 3- and 10-times the standard deviation of the blank samples (n=3). The Raschig bias is the calculated differences obtained from the line equation of the orthogonal distance regression. The determined amount of HBr (in nmol) is given for comparison with lab experiments.**

| Date | Total Br* [ppb] | HBr [ppb] | LOD [ppb] | LOQ [ppb] | Raschig bias [ppb] | HBr on denuder [nmol] | HBr/ total Br [%] | Comment |
|---|---|---|---|---|---|---|---|---|

| | | | | | | | | |
|---|---|---|---|---|---|---|---|---|
| **18.07.2016** | | | | | | | | |
| Santiago Rim | 1.85 ± 0.04 | 1.65 ± 0.05 | 0.04 | 0.12 | - 0.13 | 1.42 ± 0.04 | 89 | |
| Nindiri Rim | 1.31 ± 0.03 | 0.44 ± 0.03 | 0.02 | 0.06 | - 0.17 | 0.38 ± 0.02 | 34 | |
| **20.07.2016** | | | | | | | | |
| Santiago Rim | 1.55 ± 0.03 | 1.14 ± 0.05 | 0.07 | 0.24 | - 0.15 | 0.92 ± 0.04 | 74 | |
| Nindiri Rim | 1.22 ± 0.03 | 0.55 ± 0.05 | 0.09 | 0.29 | - 0.17 | 0.45 ± 0.04 | 45 | |
| Caldera Valley | Not available | <LOD | 1.39 | 3.99 | | 0.03 ± 0.01 | | UAV-based sampling |
| | | <LOD | 1.46 | 3.81 | | 0.02 ± 0.01 | | |
| **21.07.2016** | | | | | | | | |
| Santiago Rim | 3.05 ± 0.05 | 1.97 ± 0.11 | 0.08 | 0.27 | - 0.12 | 1.53 ± 0.09 | 65 | Simultaneous |
| | | 1.82 ± 0.10 | 0.06 | 0.26 | - 0.13 | 1.42 ± 0.08 | 60 | |
| Nindiri Rim | 1.81 ± 0.04 | 0.55 ± 0.05 | 0.05 | 0.15 | - 0.17 | 0.58 ± 0.05 | 30 | |
| | 2.56 ± 0.06 | 1.17 ± 0.07 | 0.07 | 0.23 | - 0.15 | 0.91 ± 0.05 | 46 | Simultaneous |
| | | 0.97 ± 0.09 | 0.09 | 0.30 | - 0.16 | 0.75 ± 0.07 | 38 | |

\* Total Bromine determined by Raschig Tube samples adopted from Rüdiger et al. (2021)

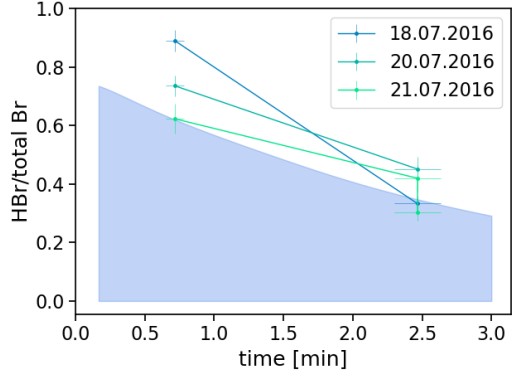

**Figure 5: Fraction of HBr (determined by EP-coated denuders) to total bromine (determined by Raschig Tubes). Assuming a windspeed of 5 m/s, HBr fractions decrease on average from 0.75 ± 0.11 at 0.7 min at Santiago Rim to 0.36 ± 0.08 at 2.5 min on Nindiri Rim. The colored area in light blue describes the fraction of HBr calculated by the model of Rüdiger et al. (2021) (model parameter to identify the selected run: An initial ratio of 10:90 of atmospheric and magmatic gas was assumed at high temperatures. The output was then quenched to 30 ppm SO$_2$ for the start of the low temperature chemistry. The proportions of hydroxyl radicals (OH), hydrogen peroxide (H$_2$O$_2$) and hydroperoxyl radicals (HO$_2$) and nitric oxide radicals (NOx) correspond to the atmospheric background composition. Within 10 minutes, dilution by a factor of 1/e (0.37) occurred. The number of particles per m$^3$ was 3 x10$^9$, their radius 300 nm.)**

## 4 Conclusion

Our goal is to be able to make a statement about the inner life of the volcano by monitoring the volcanic gases. Bromine has shown to be a potentially suitable candidate and has been correlated with changes in the activity of a volcano. Particularly attractive is the monitoring of BrO, which can be detected with remote sensing techniques. However, in order to draw conclusions about the volcanic activity by observing the volcanic bromine outgassing, we need to understand what happens to the bromine speciation after they are released in the atmosphere.

In order to get closer to this goal, a measurement method was drafted, which is able to make a certain bromine species measurable: The diffusion denuder-based method uses an internal coating that can chemically derivatize a certain species.

To make the method applicable to HBr, different coatings were tested. The coatings were based on the reactivity of appropriate epoxides towards HBr and the immobilization of HBr on the denuder walls by formation of a bromohydrin. After extraction and enrichment steps in the laboratory, this *in situ* derivatization allows the separation and detection of derivate by means of high-performance liquid chromatography-mass spectrometry.

EP was shown to be the most promising substance for the determination of the expected HBr concentrations under ground-based sampling conditions. The collection efficiency of a denuder coated with 45 µmol EP was $99.5 \pm 0.4$ % for 0.2 µmol HBr. Also, besides 5 µmol HCl, 0.2 µmol HBr could be retained to $98.1 \pm 0.4$ %. The relative standard deviation of comparable samples was 8 %.

A parallel sampling of the denuder method and Raschig Tubes revealed the comparability of both methods. For the prediction of the denuder values from the Raschig results the linear relationship with a slope of 0.97 ($\pm$ 0.10) and an intercept of 0.18 ($\pm$ 0.10) with a residual covariance of 0.25 was found. From this we concluded that we can consider measured HBr from denuders as a fraction of the total bromine given from Raschig Tubes.

The denuder sampling assembly was deployed in the volcanic plume of Masaya, Nicaragua, in July 2016. Samples were collected at the Santiago crater rim and on the Nindiri rim. Gaseous HBr in the concentration range of 0.44-1.97 ppb was detected with detection and quantification limits below 0.1 and 0.3 ppb, respectively, under typical ground-based sampling conditions. HBr contribution to total bromine decreased from an average of $75 \pm 11$ % at the Santiago rim with an estimated plume age of 0.7 min (214 m distance to the Masaya emission source) to $36 \pm 8$ % at 2.5 min at the Nindiri rim (740 m distance). The UAV-based data at advanced plume ages were below the detection limit. Especially the extension of flight time but possibly also an optimization of the method (e.g. reduction of the amount of coating reagent) may also allow UAV-based HBr measurements in the future.

The recently published model by Rüdiger et al. (2021) is based on data collected simultaneously with those shown here. A comparison shows a slightly higher trend for HBr on average than expected from the model.

The new method can now contribute to visualize the transformations of the bromine species in the volcanic plume. The extension of speciation knowledge may allow model runs to be further refined in the future. It would be a great step if we can

predict volcanic activity by monitoring volcanic bromine gases. Even better if these observations can be accomplished with BrO-monitoring by remote sensing methods.

*Data Availability.* The data used in this study is available on request from Thorsten Hoffmann (t.hoffmann@uni-mainz.de).

*Author contribution.* AG and TH designed the framework of the study. AG established the new measurement technique and carried out the experiments and data processing. AG and ML performed the analysis of Raschig Tube samples and processed the data. AG and NB collected the data sets on Masaya volcano. AG led the writing of the manuscript and all co-authors
participated in investigating and interpreting results and editing the manuscript.

*Competing interest.* The authors declare that they have no conflict of interest.

*Acknowledgements.* AG, NB and TH acknowledge support by the research center "Volcanoes and Atmosphere in Magmatic,
Open Systems" (VAMOS) at the University of Mainz, Germany and are thankful for the support by INETER (Nicaragua). The authors also thank Julian Rüdiger for his support on the collection of data sets on Masaya volcano.

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
