# Peer review of "Bromine Speciation in Volcanic Plumes: New in-situ Derivatization LC-MS Method for the Determination of Gaseous Hydrogen Bromide by Gas Diffusion Denuder Sampling"

_Atmospheric Measurement Techniques, 2020_

## Referee Comment (RC1) · Anonymous Referee #1 · 17 Dec 2020

The paper by Gutmann et al describes the evaluation and field deployment of a new technique to measure HBr in volcanic plumes via denuder sampling and subsequent analysis by LC-MS. Given the current reliance on BrO measurements obtained via remote sensing to characterize halogen chemistry in these plumes, the development of methods to measure HBr represents an important contribution to the field. While the manuscript would be improved greatly by a characterization of the sensitivity and dynamic range of this technique, the authors do demonstrate the technique is selective for HBr and that the fraction of HBr/(Total Bromine) can be obtained as a function

of distance from the emission source which is a helpful quantity even if substantial uncertainties remain as to the actual amount of HBr measured. While I believe this paper is still a useful contribution to the community, I do want to make clear that I am not convinced the authors have demonstrated an ability to quantitatively measure atmospheric HBr at the mole fractions reported. Below I provide some suggestions for improvement of the manuscript.

Line 91: Why does 250 mL/min provide "ideal" sampling efficiency?

Line 150: While I see the actual temperature programs are given in the supplement, it would be good to give the temperature ranges and total run times for the GC separations in the main text as well.

Line 239: Why this range of values? The highest value you report in Table 2 is 1.90 ppb. Are these high values atmospherically relevant? Does this relationship hold at lower mole fractions such as those you report?

Figure 2: If the goal is to compare the two techniques using the Raschig Tube measurements as an independent variable in a calibration, an orthogonal distance regression of the two sets of measurements would more clearly demonstrate the comparability of the two sets of measurements and give the reader an idea of the sensitivity of the denuder measurements. As I point out above, the calibration should be performed over an atmospherically relevant range of values.

Section 3.5: Move details of the instrument field deployment to the methods. It would be helpful to add information about how the pump was powered during sampling both on the ground and during UAV sampling to give the reader an idea of the logistical requirements. Since UAV sampling is covered, it would also be helpful to provide the approximate weight of the instrument configured for UAV deployment. Was the instrument the only payload for the UAV or was it flown with other instruments? I understand references are provided for the details of UAV campaigns and the answers may exist there, but it would be good to provide the answers to these questions in this paper.

Line 262-275: The authors haven't actually demonstrated that they can accurately quantify atmospheric HBr at low mole fractions as their laboratory studies only go from 3 to 20 ppb. Even if one assumes these data were quantitative below 3 ppb, the inclusion of actual values below LOD in Table 2 is also not appropriate, it should just be noted that the measurements were below the detection limit.

Line 282: Rephrase "This is in very good agreement with the results obtained with the model's estimations". The point of these comparisons is an evaluation of the model performance, not affirmation of your observations. The model being consistent with your observations reflects favorably on the model, but says nothing about the accuracy of your measurements as you imply here. This sentence as written implies you have doubts about the measurements. I would also add the model prediction to Fig. 5 for ease of comparison by the reader.

Fig. 5: As discussed previously, I think adding the model results referenced in the text would be helpful. The point markers should also be changed. Right now they can be confused with error bars, which is problematic. The author's should also add error bars on the ratio since they are presented in the caption. The y-axis label is not accurate. I believe what is being presented is the ratio of HBr/reactive bromine, not the ratio of HBr/ bromine atoms which is what the label says.

---

## Referee Comment (RC2) · Anonymous Referee #3 · 5 Jan 2021

General Comments

In this work, the authors present a series of synthesized epoxides to produce a selective gaseous diffusion denuder coating for the collection of HBr, specifically applied to volcanic plumes. Significant synthesis effort has been input to the production of the molecular probes, followed by laboratory and field tests for selectivity, robustness, reproducibility and suitable quality assurance and control metrics. The authors find that their best performing probe is capable of collecting HBr from groundsite locations, but not when deployed on a UAV. They compare their results to a chemical plume

model for volcano emission chemistry and find reasonable agreement to validate their methodology. Overall, the work performed here is suitable and of interest towards publication in Atmospheric Measurement Techniques pending major revisions. Foremost, the writing style of the manuscript is in dire need of improvement. The writing feels incredibly rushed and requires a significant additional time investment. It is abrupt, can be logically disorganized at times, and outright impossible to follow at others. There is an over-reliance on short paragraphs, and under-reliance on grounding in the literature, amongst other issues (see Major and Technical comments below). In addition, important aspects of the methodological work have been completed, but are missing. This include fundamental separation performance metrics for the molecular probes and reaction products, instrument detection limits, accuracy and precision assessments, range of linearity, spike and recovery summation statistics and so on. The work has already been performed to provide this critical information and the performance of the method does not appear to have critical flaws based on what is provided in the current version of the manuscript. The revised work should be re-evaluated through peer review to ensure this remains the case.

Major Comments

1. The abstract clearly demonstrates the overall symptom of unclear writing style present throughout the manuscript. The work is presented topic by topic in short paragraphs with abrupt changes. The abstract should succinctly summarize the findings of the work with clear connection between the topics. Further to this, the abstract is nearly as long as the introduction, which is the best written aspect of this work. In locations where specific changes to organization or technical language are required, they are specifically noted in the Technical Comments below. Where the authors need to independently improve clarity, a statement of 'revise for clarity' is indicated as the intention of the writing is not easily determined and/or the authors have the freedom to re-organize according to their preference.

There is a heavy reliance on conjunctive adverbs (e.g. however, therefore) in the writing

of the manuscript. In some cases, their use undermines the findings of the experiments by suggesting contrast when there is complementarity. The authors should take specific care to use these terms sparingly and accurately. Specific removal of such terms and identification of conflicting use are noted in the Technical Comments below.

2. Technical details of the methodologies from preparation of denuders, extraction and recovery of analytes, through to separation of target products, and quality assurance and quality control all have major oversights. These are numerous and critical for this work to be published in a journal focused on measurement techniques. It appears that all the necessary work has already been performed, so additional experiments are not likely required to address the Technical Comments presented in detail below.

3. A large section of the field observations needs to be moved to the methods section. The intercomparison with the DOAS data is mentioned but results are not presented. The model-measurement comparison needs to be expanded and generally rewritten for clarify.

Technical Comments

Page 1, Abstract: Rewrite after addressing all revisions to the manuscript. Focus on a concise summary of the most important achievements of this work.

Page 2, Line 45: 'are among the not negligible constituents of volcanic emissions'. Revise for clarity.

Page 2, Line 46: 'Monitoring of halogens' should specify some landmark examples.

Page 2, Line 46: 'Already' is not needed here. Delete.

Page 2, Lines 50-51: This thought is incomplete 'made BrO/SO2 a promising monitoring tool'. Why? Is it because BrO can be monitored from space? This is what the following sentence seems to imply, but it is not clear. This paragraph should also finish with a summary statement such as 'if HBr emissions decrease like those observed previously for Cl, then it may be possible to provide greater forewarning regarding volcanic eruptions through the use of remote sensing instruments. It is therefore important to have methods to determine if such reductions in HBr do, in fact, also occur.'. The link between HBr and BrO in plumes monitored from space do not materialize without providing this connection.

Further instances of this type of oversight are noted below only as 'requires connecting sentence' below so the authors may provide their own context over the reviewer's assumed intention in their writing.

Page 2, Lines 59-60: The last sentence of this paragraph is not required. Add the reference after 'hypothetical (Gutmann et al., 2018 and references therein)'.

This paragraph also requires a connecting sentence.

Page 3, Lines 69-72: Revise for clarity. The short statement 'as products bromohydrins are formed' does not stand on its own.

Page 3, Line 75: HPLC-MS should be defined at first use. Seeing as the authors deem the inclusion of ESI in the abbreviation below, a consistent term should be used throughout the manuscript.

At the end of the introduction it is fairly common to see the structure of the presented work laid out in a short list of points. It would be highly beneficial to ensure the structure of the manuscript flows as the authors desire by outlining the order in which content is going to be presented at the end of the introduction and then maintaining that structure throughout the manuscript.

Page 3, Line 79: HPLC-MS or HPLC-ESI-MS should already have been defined in the abstract or introduction above and should be used to replace all instances of 'high-performance...' henceforth.

Page 3, Lines 86-87: 'The vials containing the analytes were weighed regularly to determine output rates'. This assumes that emissions are made in the same proportion as the mixture contained in the vial, yet HCl, HBr, and H2O will all have differing permeabilities out of the vial in these mixed aqueous solutions. Was consistency between the active molecule determined by mass loss from the vial confirmed by an independent quantitative method after scrubbing the gas flow into a collection solution (e.g. ion chromatography)?

Page 3, Line 92: should 'realized' by 'made of'?

Page 3, Line 93: 'is the sampling' should be 'is sampling'

Page 4, Line 95: No comma is required after both. Delete it.

Page 4, Lines 97-100: These three sentences can be greatly simplified because the setup is clearly depicted in Figure 1. The last two sentences can be removed and the first two combined into a more concise description.

Page 4, Lines 100-106: 'Apparently', 'However', and 'Therefore' are not needed at all in this section and can be removed. The last sentence is not needed as adding '(Fig. 1b)' after 'both vessels' at Line 98 is all that is required to direct the reader to this schematic.

There is a sentence fragment of 'indicate inaccurate gas flows or fluctuate' that seems to be left over from something else that doesn't belong here either. It is worth noting here as well, that losses of strong acids to such experimental surfaces from the gas phase is well established and expected. The literature is particularly deep on this consideration for measurements of HNO3 in atmospheric chemistry field and lab setups. This work should reference such pre-existing knowledge.

Finally, the sentence currently beginning with 'Therefore' should be the first sentence in this block of information. Followed by, the sentence currently starting with 'However' and then the one with 'Apparently'. This presents the experiments much more clearly as: i) we took care to connect things in the following way, ii) this ensures comparability, and iii) is critical in order to account for losses of strong acids to experiments surfaces.

Page 4, Figure 1: The figure caption does not meet the journal guidelines. The panels do not get described on separate lines. Please refer to other manuscripts published in

AMT to ensure correct formatting of figure and table captions. It is also difficult to see where panel (a) ends and panel (b) begins. Consider placing these in a frame or at least dividing them with a solid vertical line. Typically, the lettering for panels is placed in the upper left corner of the respective panel.

Page 4, Line 112: The use of 'treatment' is not very specific. Consider revising to 'denuder preparation, extraction, and analysis'.

Page 4, Line 114: 'to avoid photochemical reactions' of what? HBr? Or photolabile HBr precursor gases, such as BrO? Or are your molecular probes susceptible to photodegradation? Please revise for clarify.

Page 4, Lines 116: Please revise for clarity. Were the '6 times of 0.5 mL' applied all at once? After each aliquot was dried? Is the drying system a custom creation or a commercial device? The use of 'therefore installed' is confusing word choice here. If it is a custom creation, a photo of the setup and a few more specific details in the Supporting Information may be of interest to others who wish to reproduce this technique or apply it to other denuder coating setups.

Page 5, Line 123: While 'educt' technically makes sense to use here, it is fairly uncommon to encounter in this field. Consider replacing.

Page 5, Lines 129-133: The use of 'elute' here is not quite correct. This term refers to the exiting of a compound from a packed column of a compound retained on a stationary phase by a mobile phase. The methodology being used here is a solvent extraction of the denuder coating and the writing should state 'extraction' instead of 'elution'.

What does 'five steps each with 2 mL of solvent' mean? Five sequential extractions, each with a volume of 2 mL for a given solvent? Was the denuder capped and inverted to ensure solvent contact with all surfaces or was the surface simply rinsed. Please clarify.

[Figure]

For the extraction efficiency assessment, how was the amount of 0.01 mmol EPBr applied to the denuder? It is not clear if the recovered amount from the extraction method was compared to a standard solution containing the same quantity of EPBr or if it was quantified via a calibrated instrumental technique. This should be specified.

Lastly, is the 'second elution step' being performed on a previously extracted denuder? Please clarify. Technically, this is an assessment of residual EPBr recovered in a second 5x2 mL extraction.

Page 5, Line 134: 'Eluates' should be 'extracts'

Page 5, Lines 134-139: This paragraph needs reorganization for clarity. The use of formic acid appears without any motivation or reasoning, flasks are mentioned for the first time and cannot be related to prior parts of the method. The internal standards used seem to have abbreviations later in the manuscript, which are not defined here. Typically, for a concentration and recovery methodology the procedure is presented as: i) internal standard identity and quantity added, ii) conditions of the concentration step, iii) analysis. This can then be followed by the obtained recovery results without the use of formic acid, then providing the rationale for its use, and finishing with the perfect recovery results obtained.

Page 5, Table 1: This belongs in Section 3.1. It would be valuable to add the chemical structures of each epoxide molecular probe, and the HBr-specific reaction product, as new columns in this table. In addition, the 'result' for EP states 'suitable', but like the 9,10-epoxystearic acid a chlorine side product is mentioned in the manuscript, but not noted here. Should that not be listed here? Perhaps the column heading should be 'Observations' instead of 'Result'?

Page 6, Lines 146-147: State that this derivatization converts to carboxylate functionality to a trimethylsilyl derivative, reducing polarity and increasing volatility. Is the second sentence here supposed to be the details of the derivatization(70 C for 90 min)? Is states 'sample storage'. Suggest revising for clarity.

Page 6, Lines 151-152: Unit notation here is incorrect and inconsistent with unit presentation requirements of this journal (use a space between values and units). The separation programs in Tables S2-S3 can be combined into a single table with subheadings for the different analytes to reduce repetition of the headers.

Page 6, Lines 165-169: No sample chromatograms of the molecular probe, the reaction product, a sample collected under contrived laboratory conditions sampling HBr, nor actual samples are shown. The tables in the SI also suggest that a reference compound only noted as 'TBA' was used in these samples. What is this and why was it necessary to use? Where matrix effects were encountered, why were dilutions not performed on the samples to attempt reducing the noted effects of: shifting retention time and broad peak widths? Were the areas of the sample peaks within the linear calibrated range of the instrumentation? The described observations suggest that the column and/or detector may have been overloaded or saturated, respectively. The mass spectra for clean standard solutions of the probe and product, as well as mass spectra obtained from the field samples to confirm these molecules despite the altered retention times are necessary to include in the supporting information.

One crucial chromatography metric missing from the assessment of all the methods tested is the calculated resolution between the probe molecule, the reaction produce, side products, the injection standard (presumable this is what 'TBA' is in Table S4), and the internal standard for recovery corrections.

The instrument linear range, detection limits, accuracy, and precision measured over the course of this work should all be presented in this section of the methodology for the quantified molecules noted in the preceding paragraph. It is not possible to ascertain from the current manuscript if samples were analyzed under reliable analytical conditions for quantitative analysis.

Page 7, Line 173: 'a significant loss of the compound was observed during sampling' How? Was this quantified by an instrumental method? What was the exact amount

lost? Everything? How was 'significance' determined statistically?

Page 7, Line 174: 'Obviously, 2,3-…. Is too volatile.' is redundant. Delete.

Page 7, Lines 174-177: Provide quantitative information to support your conclusion. There are too many ideas in a single sentence here and the authors should attempt to separate these for clarity.

Page 7, Line 178-179: 'massively' and 'however' are not needed. Delete.

Page 7, Line 183: Should 'causes low volatility' be 'reduces volatility'?

Page 7, Lines 187-188: 'Although derivatized HBr…'. Table S4 states that these molecules had different retention times, but resolutions between the peaks are not given. What is the actual analytical issue here? And if the analytes could be quantified, what does it matter if the products all have the same m/z ion in EI if you can identify the product by injecting an authentic standard? Presumably, if there are matrix effects a spike of the product through standard addition would be a viable option, but would be more labour intensive. This discussion should be expanded and be more thorough.

Page 7, Lines 189-190: Certainly more discussion and the criteria for meeting the designation 'suitable' are warranted here!

Page 7, Line 190: 'chapters' should be 'sections'.

Page 7, Section 3.2: There are many different topics within this section that would greatly benefit from separate subheadings (e.g. Coating Breakthrough Experiments, QA/QC and Matrix Effects, Coating Stability, and Interferences and Cross Reactions). It would help organize the section and reduce the number of places where connecting sentences would otherwise be required to make the discussion contents easier to follow.

Page 7, Line 196: 'Therefore, the' can be 'We tested'. The rest of this sentence requires further clarification. One can eventually discern that the point is to apply an upper limit

of HBr mixing ratio to the denuder, based on prior ambient observations of plumes. Suggest revision for clarity.

Page 7, Line 199: How was the coating quantity on the denuders determined? Was this quantified using the instrumental methods presented or was this determined by calculation? From the description of the denuder coating section, it does not seem likely that all of the analyte solution applied to the denuder surface is retained and that likely a large amount of the applied solution exits the denuder when it is rotating on the 10 degree sloped setup with gas flowing over it. This contrast between theoretical amount of probe applied compared to the practical result of the method would be valuable to discuss here if the applied amount was not quantified.

Page 8, Line 200: 'clearly' is not required. Delete.

Page 8, Line 201: 'was below 1 %' in which denuder? The third one? For all field samples collected were both denuders extracted to ensure accurate quantitative collection of the targeted HBr? The following use of 'however' can be deleted.

Page 8, Line 202: Given the issues encountered with peak width and shape in the HPLC separation, perhaps such aggressive concentration steps were not required to retain suitable detection limits for the method? It would be worthwhile to comment on such findings here. The use of 'finally' at the end of this sentence does not really work. Consider using 'We determine that use of 15 mmol/L EP coating solution was ideal to coat denuders with 45 umol EP.'

Page 8, Lines 204-206: 'The influence of EP' due to what? Its much higher concentration relative to EPBr? What are the 450 mmol/L 'concentrations'? Are these standard solutions? Were the mixtures done at equal amounts of EP compared to EPBr? A retention effect of EP on EPBr is likely to be muted under equimolar mass loadings on the analytical column and observed more realistically by simulating ratios you'd expect to find in real samples. What does the 'n=6' refer to here? Six standard mixtures of equimolar concentration between the upper and lower bounds stated? Six injections of

the two concentrations? Does the upper limit of this range correspond to the maximum concentration of EP that could be extracted from a denuder? None of these important details are discussed, but they have surely been considered. Please expand the discussion to improve clarity and communicate the care and detail considered in this work.

What values were the standard deviation percentages determined from? Some metrics are very useful (e.g. accuracy and precision in quantitation) while others are not (e.g. retention time). This is critical to specify and justify why the metric you selected has utility in concluding that no matrix effects exist in the separation technique from the probe and product molecules.

Page 8, Line 207: 'Constricted' should be 'concentrated'. What are the 'samples' mentioned here? Contrived samples made from pure compounds, field samples, lab samples, or all of the above? How many times was the sample reanalysis performed and at what temporal frequency (e.g. monthly, weekly)? A plot of the data and clear indication of the identity of the samples is required to ascertain whether the stated stability of the probe is a sound conclusion. The trend of sample stability/degradation quantified should be presented with a figure in the Supporting Information.

Page 8, Lines 211-215: The authors present method detection and quantitation limits, but instrument performance metrics are missing from the methods section (as well as how they were determined). Since the molecular probes are novel analytes without pre-existing methods published in the literature, such QA/QC is critical to present here. Please add, so that the results presented here can be evaluated with that information in mind.

Were there detectable peaks for the reaction product in the field blanks that set the method detection limits higher than the instrument detection limits? How many field blanks were analyzed to calculate these values? What was the variance between the blanks, if any? Were there detectable reaction product peaks in a denuder that was

extracted immediately after drying? How did these compare and what does this tell you about the potential for contamination of denuders during transport? This helps clearly set up the reasoning behind the assessment of interferences and cross-reactivity that follows.

Page 8, Line 217: 'would be at least conceivable as a' is very confusing. Consider 'is the most likely' instead. 'However' at the end of this line is, again, unnecessary.

Page 8, Lines 217-218: What mixing rations of Br2 were used, what other gases were present with the Br2? This sentence requires revision for clarity and justification for the representativeness of the Br2 quantities investigated.

Page 8, Lines 219-221: Since methyl bromide was not explicitly tested, has not been detected in volcano plumes, and will not produce the same product as HBr, it is not worth discussing. Delete.

Page 8, Line 228-230: Reorganize and clarify.

Page 8, Lines 230-231: How long as the 4 ppm HCl exposure performed for? A comparable duration to field sampling volume and duration? What about integrated exposure with some approximated limits based on the literature?

Page 9, Lines 238-240: The alkaline traps are the established techniques and the new EP probe should be compared against them. Here, the comparison is made the other way around.

Page 9, Figure 3: This should be converted to a table with the +/- values given and statistical tests for similarity performed to demonstrate that both methods are measuring the same quantity of HBr. This is a more robust validation of the developed EP molecular probe. The last sentence in the caption is a repetition of text already presented in the discussion. Delete.

Page 9, Section 3.5: Much of the information presented here should be relocated to the methods section to describe the sampling site and sampling approach. Only results

and discussion of the new denuder method should be presented here.

Page 9, Lines 251 and 256-257: The UAV system needs to be briefly summarized here. Referring to prior reports alone is not adequate.

Page 9, Line 255: The NOVAC station is not defined anywhere? What does this stand for? Figure 4 does not denote it specifically. In the subsequent discussion, there is no comparison made to the DOAS measurements noted here. This is a highly valuable field intercomparison. What were the results or why was it not possible to get usable data? Ideally, this should be expanded.

Page 10, Figure 4: Color code the ground versus UAV locations. Place distances from the HBr emission source to the sampling locations on the map and remove from Table 2.

Page 10, Line 262: The range is from '< LOD' to 1.97 ppb. A reference to Table 2 should be made at the end of this sentence and the last sentence of this paragraph deleted.

Page 10, Lines 264-265: This information belongs in the caption for Table 2. Relocate.

Page 10, Line 268: The 2x2 information in brackets should be deleted and a column for the number of samples collected at each location added to Table 2.

Page 10, Line 268-270: are these mixing ratios statistically the same or different? Can the precision of the method be approximated despite the small number of replicate samples? This is an example of where determination of the method precision for the laboratory experiments can bolster the robustness of the method when applied in the field and the discussion here can bridge those findings together.

Page 10, Line 270: A deviation of a deviation is not very meaningful and a good reason for calculating it is not given. Justify or remove.

Page 10, Line 276-282: The model-measurement comparison is too brief and needs

to be expanded to clearly demonstrate the quality of the field measurements. For example, the loss of HBr in the model is stated as a percentage, but the loss of HBr observed is discussed in the same context. One has to hope they identify the correct information in Table 2 and then calculate this independently to ascertain the statemen of 'very good agreement' is true. Overall, I cannot follow the logic of this section and it weakens the manuscript when it should be strengthening it. Please revise.

Page 11, Table 2: Samples with HBr quantities below the method detection limits should be reported as '<LOD' or as '<XX' for the corresponding detection limit of a given sample. A column for the number of samples at each location and date needs to be added, as well as for the sampling method (ground vs UAV). Is the Br/S in the table measured or modeled? This needs to be stated in the figure caption and in the discussion.

Page 11, Figure 5: Can the fraction of HBr/Br from the model be added to the figure for comparison? Based on the discussion this should be possible and compare well.

Conclusions: Rewrite based on revised manuscript and in light of all comments above.

---

## Author Comment (AC1) · 3 Jul 2021

We thank the reviewer for his time to review this manuscript and the many suggestions for improvement.

Reviewer: General

The paper by Gutmann et al describes the evaluation and field deployment of a new technique to measure HBr in volcanic plumes via denuder sampling and subsequent
analysis by LC-MS. Given the current reliance on BrO measurements obtained via remote sensing to characterize halogen chemistry in these plumes, the development of methods to measure HBr represents an important contribution to the field.

Method description

While the manuscript would be improved greatly by a characterization of the sensitivity and dynamic range of this technique, the authors do demonstrate the technique is selective for HBr and that the fraction of HBr/(Total Bromine) can be obtained as a function of distance from the emission source which is a helpful quantity even if substantial uncertainties remain as to the actual amount of HBr measured.

While I believe this paper is still a useful contribution to the community, I do want to make clear that I am not convinced the authors have demonstrated an ability to quantitatively measure atmospheric HBr at the mole fractions reported. Below I provide some suggestions for improvement of the manuscript.

Response: There is indeed a large amount of uncertainty in many of these measurements, and we would like to address and discuss them. From the comments on the review, we also recognized that parts of the discussion were missing or could be developed further, and we have expanded or reworded the text as needed. Please see the details for the changes in the responses to the following comments below.

Reviewer: Line 91: Why does 250 mL/min provide "ideal" sampling efficiency?

Response: One influence on the choice was that an optimal flow rate of 250 ml/min was determined for a previously developed denuder system for reactive bromine species (Rüdiger et al., 2017). The two denuder systems are supposed to complement each other and, in the best case, sample in parallel with the same pump. Even though a higher flow rate would be advantageous because more sample volume can be enriched, it is a logistical advantage if both systems sample at the same flow rate.

It was described in the introduction that under the given conditions with a flow rate of

250 ml/min, the probability of gaseous HBr molecules diffusing to the denuder walls and reacting with the coating is 99.999997%. With the breakthrough studies in section 3.2.1, it has also been practically proven that a slower flow is not necessary.

Nevertheless, it is true that the wording 'ideal' is misleading here and has been changed.

Changes: A sampling flow rate of 250 mL/min was used.

Reviewer: Line 150: While I see the actual temperature programs are given in the supplement, it would be good to give the temperature ranges and total run times for the GC separations in the main text as well.

Changes: The oven temperature programs (range from 90-300 °C with total runtimes of 9-30 min) and the ions used for quantification are summarized in the supplementary material (Table S1).

Reviewer: Line 239: Why this range of values? The highest value you report in Table 2 is 1.90ppb. Are these high values atmospherically relevant? Does this relationship hold at lower mole fractions such as those you report?

Response: This comment highlights a clear gap in the comprehensibility of the experiments. The presentation of the test gas concentration alone was misleading here. Test gas concentrations were higher than expected in the atmosphere but sampling duration was shorter, resulting in a sampled amount of HBr in the same range as the field samples. For a better comparability the amounts of HBr (in nmol) were added to the laboratory studies and field sample results.

Still, this will only reflect the higher concentrated data from the set shown here from Masaya, that's true. For upcoming samples from volcanic plumes where we expect higher bromine emissions, such as from Etna, the values may fit better.

Changes: Modified Table 3 and 4

Reviewer: Figure 2: If the goal is to compare the two techniques using the Raschig Tube measurements as an independent variable in a calibration, an orthogonal distance regression of the two sets of measurements would more clearly demonstrate the comparability of the two sets of measurements and give the reader an idea of the sensitivity of the denuder measurements. As I point out above, the calibration should be performed over an atmospherically relevant range of values.

Response: Yes, the comparison with the approved alkaline method was used to prove the accuracy and, above all, the comparability of both methods, since a ratio is to be formed from their results.

We acknowledge the reviewer's comment about the need to discuss the comparison between the methods more thoroughly. We followed the suggestion to perform an orthogonal distance regression for comparing the results of both methods.

Changes: New Figure 4 (Fig. 1). On average, the results related to denuder sampling yielded 99 ± 11 % of the HBr values determined by the Raschig Tube. An orthogonal distance regression was performed and is shown in Fig. 4. Based on the line equation obtained, small values, such as those observed here in the field samples, can yield higher results from denuder determinations than expected via the Raschig Tube. We have concluded, that the HBr values determined by denuders in field samples can be considered a fraction of the total bromine determined by the Raschig Tubes. To account for the comparison studied, the deviation found is included as an error of the denuder field samples in Table 4.

Reviewer: Section 3.5: Move details of the instrument field deployment to the methods. It would be helpful to add information about how the pump was powered during sampling both on the ground and during UAV sampling to give the reader an idea of the logistical requirements. Since UAV sampling is covered, it would also be helpful to provide the approximate weight of the instrument configured for UAV deployment. Was the instrument the only payload for the UAV or was it flown with other instruments? I

understand references are provided for the details of UAV campaigns and the answers may exist there, but it would be good to provide the answers to these questions in this paper.

Response: We agree with the reviewer that this additional information makes the manuscript easier to follow. The description of the fieldwork has been moved to the new section '2.4 Field application at Masaya 2016'. A brief description of the drone setups used has also been included here. The GilAir Plus used in ground-based sampling is an air sampling pump, commonly used for person-carried sampling of hazards at work. The pump already contains its own battery and weighs approximately 600 g. In the manuscript, we have given the reference to the size and the included battery.

Changes: 2.4 Field application at Masaya 2016 A first set of field samples was collected between 18.-21. of July 2016 at the Santiago Crater of the Masaya volcano (Nicaragua). A detailed description of the location can be found in Rüdiger et al. (2021). In summary, sets from different methods were collected simultaneously together at changing locations with various distances (200-2000 m, Fig. 3b) to Masaya's emission source at Santiago crater (Fig. 3b).

A total of eight ground-based and two UAV-based samples for the newly developed denuder method are presented here. In ground-based sampling sets, two denuders were sampled in series (Fig. 3a). Both denuders were extracted and results were summarized. Sampling was performed by a Gilian GilAir Plus handheld pump (battery included) with a flow rate of 250 ml/min for about 1-1.5 hours for each denuder. In addition to EP-coated denuders, samples with 1,3,5-trimethoxybenzene coated denuders for the determination of reactive bromine. as well as Raschig Tubes as alkaline traps for the determination of total bromine and total sulfur were collected simultaneously side by side. The results of these samples can also be found in Rüdiger et al. (2021).

First drone-based samples were collected with an UAV using a small four-rotor multicopter with foldable arms (Black Snapper, Globe Flight, Germany) called RAVEN (Rüdiger et al., 2018). For the UAV-based sampling, a remotely controlled sampler (called Black Box) was used and is also described in detail in Rüdiger et al. (2018). The Black Box enabled logging of the sampling duration and $SO_2$ mixing ratios via the built-in $SO_2$ electrochemical sensor (CiTiceL 3MST/F, City Technology, Portsmouth, United Kingdom). The Black Box has 20x14x13 cm. With this setup (Black Box + denuder) of approx. 1 kg we achieved flight times of up to 15 min. In drone-based sampling flights, individual denuders were used with sampling times between 5-10 minutes.

Reviewer: Line 262-275: The authors haven't actually demonstrated that they can accurately quantify atmospheric HBr at low mole fractions as their laboratory studies only go from 3 to 20 ppb. Even if one assumes these data were quantitative below 3 ppb, the inclusion of actual values below LOD in Table 2 is also not appropriate, it should just be noted that the measurements were below the detection limit.

Response: The revised tables now show that the laboratory studies are at least in the range of the samples found and cover the larger values. Values below the LOD have been marked as such.

Changes: Modified Tables 3 and 4.

Reviewer: Line 282: Rephrase "This is in very good agreement with the results obtained with the model's estimations". The point of these comparisons is an evaluation of the model performance, not affirmation of your observations. The model being consistent with your observations reflects favorably on the model, but says nothing about the accuracy of your measurements as you imply here. This sentence as written implies you have doubts about the measurements. I would also add the model prediction to Fig. 5 for ease of comparison by the reader.

Response: We agree with the reviewer's suggestions that the statements had to be rephrased. In addition, the model run mentioned was taken over into the figure

Changes: Modified Figure 5 (Fig. 2).

In the work of Rüdiger et al. (2021) the results of total bromine, total reactive bromine and bromine monoxide from accompanying methods were used to run the atmospheric box model CAABA/MECCA, which was initialized by a high-temperature equilibrium model. The model run that best described the data in Rüdiger et al. (2021) was used here for comparison and is highlighted in light blue for the ratio of HBr/total Br in Fig. 5. This run was based on a Br/S ratio of 7.4 x10 4. The Br/S ratio for the measurements considered here on 18.-21.7. was on averaged $6.2 \pm 1.0$ x10 4. Even though the general trend between measured values and model predictions is consistent, on average the measured values appear to be slightly higher than those calculated by this model run. Following the observations in Rüdiger et al. (2021), a cause may be the influence of aerosol. Aerosol was not measured simultaneously, smaller particle number concentrations and diameters than assumed may lead to slower HBr loss than expected. Also, deviation from the assumed wind speed can lead to a horizontal shift of the measurements while the deviations between Denuder and Raschig method cause a vertical shift. Overall, the trend observed would have to be confirmed by further samples. These samples give us a first idea that we can confirm our general idea about the HBr consumption. Of course, a solid foundation will require many field samples and further consideration of the two methods used and their joint application at the expected concentrations.

Reviewer: Fig. 5: As discussed previously, I think adding the model results referenced in the text would be helpful. The point markers should also be changed. Right now they can be confused with error bars, which is problematic. The author's should also add error bars on the ratio since they are presented in the caption. The y-axis label is not accurate. I believe what is being presented is the ratio of HBr/reactive bromine, not the ratio of HBr/ bromine atoms which is what the label says

Response: We liked the suggestion to add the model run to the figure. It seems that the choice to add caps at the end of the error bars was confusing. They have now been removed to make the figure clearer. The y-axis labels have also been adjusted. What

can be seen here is the ratio of HBr to total bromine, which was labeled only as Br, misleading to bromine radicals. We changed to 'total Br' for clarification.

Changes: Modified Figure 5.

Please also note the supplement to this comment:
https://amt.copernicus.org/preprints/amt-2020-357/amt-2020-357-AC1-supplement.pdf

[Figure]

**Fig. 1.** = Figure 4: Figure 4: Denuder results are plotted against Raschig results. The orthogonal distance regression model resulted in: y = 0.97 ($\pm$ 0.10) * x + 0.18 ($\pm$ 0.10) and a residual variance of 0.25.

[Figure]

**Fig. 2.** = Figure 5: Fraction of HBr (determined by EP-coated denuders) to total bromine (determined by Raschig Tubes). Assuming a windspeed of 5 m/s, HBr fractions decrease on average from $0.75 \pm 0.11$ at 0.7

**Supplement:**

**Modified tables in response to Review 1**

**Table 3: Comparison of simultaneously test gas sampling of EP coated denuders and Raschig Tubes for gaseous HBr**
5   **determination. The setup is shown in Fig. 1b.**

| Experiment | Denuder | | Raschig | |
|---|---|---|---|---|
| | Sampled amount of HBr [nmol] | Calculated HBr in sampled test gas [ppb] | Sampled amount of HBr [nmol] | Calculated HBr in sampled test gas [ppb] |
| 1 | $1.8 \pm 0.6$ | $3.6 \pm 1.6$ | $26.0 \pm 3.6$ | $4.1 \pm 0.6$ |
| 2 | $2.3 \pm 0.7$ | $5.8 \pm 2.2$ | $28.1 \pm 3.9$ | $5.1 \pm 0.7$ |
| 3 | $3.0 \pm 0.4$ | $7.7 \pm 1.2$ | $41.6 \pm 5.3$ | $7.4 \pm 1.0$ |
| 4 | $4.8 \pm 0.7$ | $14.4 \pm 2.3$ | $83.0 \pm 11.7$ | $16.8 \pm 2.4$ |
| 5 | $7.6 \pm 0.7$ | $18.5 \pm 1.8$ | $111.7 \pm 14.7$ | $17.9 \pm 2.4$ |

**Table 4: Results of denuder measurements sampled in Masaya's plume on three days in July 2016. Sampling has been performed at three different locations with the following distances to the emission source: Santiago Rim 215 ± 50 m, Nindiri Rim 740 ± 50 m and in the Caldera Valley 2000 ± 150 m (Fig. 3b). Total Bromine has been determined by simultaneously applied Raschig Tubes (details in Rüdiger et al. 2021). HBr concentrations (in ppb) were determined by EP-coated denuders. Their respective LOD and LOQ were calculated based on the signal-to-noise approach using 3- and 10-times the standard deviation of the blank samples (n=3). The Raschig bias is the calculated differences obtained from the line equation of the orthogonal distance regression. The determined amount of HBr (in nmol) is given for comparison with lab experiments.**

| Date | Total Br* [ppb] | HBr [ppb] | LOD [ppb] | LOQ [ppb] | Raschig bias [ppb] | HBr on denuder [nmol] | HBr/ total Br [%] | Comment |
|---|---|---|---|---|---|---|---|---|
| 18.07.2016 | | | | | | | | |
| Santiago Rim | 1.85 ± 0.04 | 1.65 ± 0.05 | 0.04 | 0.12 | - 0.13 | 1.42 ± 0.04 | 89 | |
| Nindiri Rim | 1.31 ± 0.03 | 0.44 ± 0.03 | 0.02 | 0.06 | - 0.17 | 0.38 ± 0.02 | 34 | |
| 20.07.2016 | | | | | | | | |
| Santiago Rim | 1.55 ± 0.03 | 1.14 ± 0.05 | 0.07 | 0.24 | - 0.15 | 0.92 ± 0.04 | 74 | |
| Nindiri Rim | 1.22 ± 0.03 | 0.55 ± 0.05 | 0.09 | 0.29 | - 0.17 | 0.45 ± 0.04 | 45 | |
| Caldera Valley | Not available | <LOD | 1.39 | 3.99 | | 0.03 ± 0.01 | | UAV-based sampling |
| | | <LOD | 1.46 | 3.81 | | 0.02 ± 0.01 | | |
| 21.07.2016 | | | | | | | | |
| Santiago Rim | 3.05 ± 0.05 | 1.97 ± 0.11 | 0.08 | 0.27 | - 0.12 | 1.53 ± 0.09 | 65 | Simultaneous |
| | | 1.82 ± 0.10 | 0.06 | 0.26 | - 0.13 | 1.42 ± 0.08 | 60 | |
| Nindiri Rim | 1.81 ± 0.04 | 0.55 ± 0.05 | 0.05 | 0.15 | - 0.17 | 0.58 ± 0.05 | 30 | |
| | 2.56 ± 0.06 | 1.17 ± 0.07 | 0.07 | 0.23 | - 0.15 | 0.91 ± 0.05 | 46 | Simultaneous |
| | | 0.97 ± 0.09 | 0.09 | 0.30 | - 0.16 | 0.75 ± 0.07 | 38 | |

* Total Bromine determined by Raschig Tube samples adopted from Rüdiger et al. (2021)

---

## Author Comment (AC2) · 3 Jul 2021

We thank the reviewer very much for his effort to provide such a detailed review of the manuscript and the many suggestions for improvement.

General Comments

Reviewer: In this work, the authors present a series of synthesized epoxides to produce a selective gaseous diffusion denuder coating for the collection of HBr, specifically applied to volcanic plumes. Significant synthesis effort has been input to the production of the molecular probes, followed by laboratory and field tests for selectivity, robustness, reproducibility and suitable quality assurance and control metrics. The authors find that their best performing probe is capable of collecting HBr from groundsite locations, but not when deployed on a UAV. They compare their results to a chemical plume model for volcano emission chemistry and find reasonable agreement to validate their methodology. Overall, the work performed here is suitable and of interest towards publication in Atmospheric Measurement Techniques pending major revisions. Foremost, the writing style of the manuscript is in dire need of improvement. The writing feels incredibly rushed and requires a significant additional time investment. It is abrupt, can be logically disorganized at times, and outright impossible to follow at others. There is an over-reliance on short paragraphs, and under-reliance on grounding in the literature, amongst other issues (see Major and Technical comments below). In addition, important aspects of the methodological work have been completed, but are missing. This include fundamental separation performance metrics for the molecular probes and reaction products, instrument detection limits, accuracy and precision assessments, range of linearity, spike and recovery summation statistics and so on. The work has already been performed to provide this critical information and the performance of the method does not appear to have critical flaws based on what is provided in the cur-rent version of the manuscript. The revised work should be re-evaluated through peer review to ensure this remains the case.

Response: We agree with the reviewer's suggestions, which encouraged us to revise the entire manuscript. From the comments on the review, we also recognized that parts of the discussion were missing or could be developed further, and we have expanded or reworded the text as needed. Please see the details for the changes in the responses to the following comments below.

Major Comments

1. Reviewer: The abstract clearly demonstrates the overall symptom of unclear writing

style present throughout the manuscript. The work is presented topic by topic in short para-graphs with abrupt changes. The abstract should succinctly summarize the findings of the work with clear connection between the topics. Further to this, the abstract is nearly as long as the introduction, which is the best written aspect of this work.

In locations where specific changes to organization or technical language are required, they are specifically noted in the Technical Comments below. Where the authors need to independently improve clarity, a statement of 'revise for clarity' is indicated as the intention of the writing is not easily determined and/or the authors have the freedom to reorganize according to their preference.

There is a heavy reliance on conjunctive adverbs (e.g. however, therefore) in the writing of the manuscript. In some cases, their use undermines the findings of the experiments by suggesting contrast when there is complementarity. The authors should take specific care to use these terms sparingly and accurately. Specific removal of such terms and identification of conflicting use are noted in the Technical Comments below.

Response: The abstract has been rewritten and the comments given were used as a starting point, the manuscript was further improved beyond that where it seemed necessary. Please refer to the following answers for detailed changes.

2. Technical details of the methodologies from preparation of denuders, extraction and recovery of analytes, through to separation of target products, and quality assurance and quality control all have major oversights. These are numerous and critical for this work to be published in a journal focused on measurement techniques. It appears that all the necessary work has already been performed, so additional experiments are not likely required to address the Technical Comments presented in detail below.

Response: The technical details have been expanded in the method section, the denuder performance in section 3 and more details have been added to the SI. Please see the details in the following responses.

3. A large section of the field observations needs to be moved to the methods section. The intercomparison with the DOAS data is mentioned but results are not presented. The model-measurement comparison needs to be expanded and generally rewritten for clarify.

Response: The manuscript structure and model discussion were revised and restructured.

Since the BrO data obtained by DOAS are already processed in the model run used here for comparison and the data do not otherwise provide any further means of comparison, they have been removed from this manuscript.

Technical Comments

Reviewer: Page 1, Abstract: Rewrite after addressing all revisions to the manuscript. Focus on a concise summary of the most important achievements of this work.

Response: We acknowledge the reviewer's comment about the need to focus the text. The abstract has been rewritten. Special attention was paid to concentrate on the most important points of this paper.

Changes: Abstract. The chemical characterization of volcanic gas emissions gives insights into the interior of volcanoes. Bromine species have been correlated with changes in the activity of a volcano. In order to exploit the volcanic bromine gases, we need to understand what happens to them after they are outgassed into the atmosphere.

This study aims to shed light on the conversion of bromospecies after degassing. The method presented here allows the specific analysis of gaseous hydrogen bromide (HBr) in volcanic environments. HBr is immobilized by reaction with 5,6-epoxy-5,6-dihydro-[1,10]-phenanthroline (EP), which acts as an inner coating inside of diffusion denuder tubes (in situ derivatization). The derivative is analyzed by high-performance liquid chromatography coupled to electrospray ionization mass spectrometry (HPLC-

MS).

The collection efficiency for HBr (99.5 %), collection efficiency for HBr alongside HCl (98.1%) and the relative standard deviation of comparable samples (8 %) have been investigated. The comparison of the new denuder-based method and Raschig Tubes as alkaline trap resulted on average in a relative bias between both methods of $10 \pm 6$ %.

The denuder sampling setup was applied in the plume of Masaya (Nicaragua) in 2016. HBr concentrations in the range between 0.44 and 2.27 ppb were measured with limits of detection and quantification below 0.1 and 0.3 ppb respectively. The relative contribution of HBr as a fraction of total bromine decreased from $75 \pm 11$ % at Santiago rim (214 m distance to the volcanic emission source) to $36 \pm 8$ % on Nindiri rim (740 m distance).

A comparison between our data and the previously calculated HBr, based on the CAABA/MECCA box model, showed a slightly higher trend for the HBr fraction on average than expected from the model. Data gained from this new method can further refine model runs in the future.

Reviewer: Page 2, Line 45: 'are among the not negligible constituents of volcanic emissions'. Revise for clarity.

Changes: Besides H2O, CO2 and sulfur species, also halogens are constituents of volcanic emissions e.g. in arc volcanoes 0.84% HCl, 0.061% HF, 0.0025% HBr (Textor et al., 2004; Gerlach, 2004).

Reviewer: Page 2, Line 46: 'Monitoring of halogens' should specify some landmark examples.

Response: Later in this paragraph, examples are already given for the monitoring of halogens and their correlation with volcanic activity:

- Noguchi und Kamiya, 1963

For BrO:

- Lübcke et al., 2014

- Hörmann et al., 2013

You can see the examples mentioned in the comment after next.

Reviewer: Page 2, Line 46: 'Already' is not needed here. Delete.

Response: Done.

Reviewer: Page 2, Lines 50-51: This thought is incomplete 'made BrO/SO2 a promising monitoring tool'. Why? Is it because BrO can be monitored from space? This is what the following sentence seems to imply, but it is not clear. This paragraph should also finish with a summary statement such as 'if HBr emissions decrease like those observed previously for Cl, then it may be possible to provide greater forewarning regarding volcanic eruptions through the use of remote sensing instruments. It is therefore important to have methods to determine if such reductions in HBr do, in fact, also occur.'. The link between HBr and BrO in plumes monitored from space do not materialize without providing this connection. Further instances of this type of oversight are noted below only as 'requires connecting sentence' below so the authors may provide their own context over the reviewer's assumed intention in their writing.

Response: Done.

Changes: . . . For example, Noguchi and Kamiya determined the plume composition of Mt. Asama over months and observed a decreasing Cl/S ratio before an eruption in 1958 (Noguchi and Kamiya, 1963). The discovery of bromine monoxide (BrO) in volcanic plumes and the observed correlation between the simultaneously determined BrO/SO2 ratio and volcanic activity by automated instruments (e.g. Lübcke et al., 2014) made BrO/SO2 a promising monitoring tool. A major advantage of BrO is typically negligible background in the atmosphere, concluding that findings clearly derive from magma degassing. The BrO/SO2 ratio is measurable simultaneously by

remote sensing instruments which facilitates the applicability for monitoring volcanic activity compared to in situ sampling techniques. Even inaccessible volcanoes can be observed with satellite-based remote sensing instruments (e.g. Hörmann et al., 2013). If If the informative value of Br emissions about volcanic activity is confirmed„ then it may be possible to provide greater forewarning regarding volcanic eruptions via BrO monitoring through the use of remote sensing instruments.

It is therefore important to have methods to determine how bromine species are transformed in volcanic plumes. . . .

Reviewer: Page 2, Lines 59-60: The last sentence of this paragraph is not required. Add the reference after 'hypothetical (Gutmann et al., 2018 and references therein)'.

This paragraph also requires a connecting sentence.

Response: Done.

Changes: . . . However, due to the lack of applicable measurement techniques for individual bromine species, comparison of expected bromine species conversion rates with measurements are still hypothetical (Gutmann et al., 2018 and references therein).

A promising approach for the revelation of the bromine speciation are gas diffusion denuder techniques. . . .

Reviewer: Page 3, Lines 69-72: Revise for clarity. The short statement 'as products bromohydrins are formed' does not stand on its own.

Response: The sentence has been revised. The ongoing reaction has already been described the sentence before.

Changes: Epoxides are effective reagents for a rapid reaction with gaseous HBr, since they show acid-catalyzed ring-opening reactions with nucleophilic reagents, in the case of HBr both properties being present in one molecule. Therefore, as products from the reaction of epoxides with HBr bromohydrins are formed (Becker and Beckert, 2004).

Reviewer: Page 3, Line 75: HPLC-MS should be defined at first use. Seeing as the authors deem the inclusion of ESI in the abbreviation below, a consistent term should be used throughout the manuscript. At the end of the introduction it is fairly common to see the structure of the presented work laid out in a short list of points. It would be highly beneficial to ensure the structure of the manuscript flows as the authors desire by outlining the order in which content is going to be presented at the end of the introduction and then maintaining that structure throughout the manuscript.

Response: HPLC-ESI-MS has now been used throughout the manuscript and a paragraph has been added on the expected course of the manuscript.

Changes: In this study, the aim was to develop a measurement technique with which it is possible to detect gaseous HBr in volcanic gases. Enrichment of gaseous HBr by chemical bonding to a coating in denuders seemed the most promising method. First, a source of HBr test gas was developed, which is included in the methods section. Followed by the experimental setup, handling of the denuders, analysis of the samples, and finally the description of the application of the method in Masaya's plume in 2016. After testing five different coatings, 5,6-epoxy-5,6-dihydro-[1,10]-phenanthroline (EP) showed the best conditions for a successful method development. The performance of the EP-coated denuders is described in more detail in the following. In order to see to what extent a supplementation of simultaneously collected field samples with other methods is possible, a comparative measurement of the new denuder method was compared with alkaline samples from a Raschig tube. Finally, the results of the new method applied in the volcanic plume of Masaya volcano are presented.

Reviewer: Page 3, Line 79: HPLC-MS or HPLC-ESI-MS should already have been defined in the abstract or introduction above and should be used to replace all instances of 'high-performance...' henceforth.

Response: Done.

Reviewer: Page 3, Lines 86-87: 'The vials containing the analytes were weighed reg-

ularly to determine output rates'. This assumes that emissions are made in the same proportion as the mixture contained in the vial, yet HCl, HBr, and H2O will all have differing permeabilities out of the vial in these mixed aqueous solutions. Was consistency between the active molecule determined by mass loss from the vial confirmed by an independent quantitative method after scrubbing the gas flow into a collection solution (e.g. ionchromatography)?

Response: HBr and water form an azeotrope at 48% HBr. Therefore, it may be expected that the ratio in the gas phase is near that given in the solution.

It seems that the reviewer also assumes that HBr and HCl were provided in the same vial. Therefore, we would like to clarify that both acids were deposited in separate vials. We have also clarified this in the text. Further, we have added the output rates of the vials as a figure to the Supplementary.

The output rate from the source has not been confirmed by other methods. That's why the accuracy of the denuder method has been performed through the intercomparing with Raschig Tubes.

Changes:

2.1.1 Test gas sources

A diffusion gas source using a brown glass vial with septum cap as described in Rüdiger et al. (2017) was filled with 48 % aqueous HBr and stored under nitrogen flow at 30 or 50° C (Fig. 1a). The incoming gas stream was thermostated before it reached the diffusion source. The vials containing the analytes were weighed regularly to determine output rates. The length and diameter of the capillary, as well as the temperature, controlled the output rate. A 5 cm capillary with 0.32 mm inner diameter was used for the HBr source. Taking advantage of the azeotropic behaviour of the HBr-water-mixture (Haase et al., 1963) we observed a constant output rate for HBr of 108 $\mu$g/d and 415 $\mu$g/d for 30 and 50 °C respectively (Fig. S2).

For experiments testing the collection efficiency of HBr aside hydrogen chloride (HCl), a vial following the same procedure using a 2 cm capillary with 0.64 mm inner diameter was prepared containing 30 % aqueous HCl. We observed an output rate for HCl of 4.33 mg/day at 50 °C. For the experiments both vials (one each for HBr and HCl) were stored side by side in the test gas storage vessel (Fig 1a).

For Br2, the permeation source described in Rüdiger et al. (2017) was used with an output rate of 775 $\mu$g/d.

Reviewer: Page 3, Line 92: should 'realized' by 'made of'?

Response: Yes, has been changed.

Reviewer: Page 3, Line 93: 'is the sampling' should be 'is sampling'

Response: Done.

Reviewer: Page 4, Line 95: No comma is required after both. Delete it.

Response: Done.

Reviewer: Page 4, Lines 97-100: These three sentences can be greatly simplified because the setup is clearly depicted in Figure 1. The last two sentences can be removed and the first two combined into a more concise description.

Response: The setup description was shortened to the following:

Changes: The target HBr concentrations were generated by a standard gas bottle with 102.8 $\pm$ 3.1 ppm HBr diluted with 3-neck flasks.

Reviewer: Page 4, Lines 100-106: 'Apparently', 'However', and 'Therefore' are not needed at all in this section and can be removed. The last sentence is not needed as adding '(Fig. 1b) 'after 'both vessels' at Line 98 is all that is required to direct the reader to this schematic.

There is a sentence fragment of 'indicate inaccurate gas flows or fluctuate' that seems

to be left over from something else that doesn't belong here either. It is worth noting here as well, that losses of strong acids to such experimental surfaces from the gasphase is well established and expected. The literature is particularly deep on this consideration for measurements of HNO3 in atmospheric chemistry field and lab setups. This work should reference such pre-existing knowledge.

Finally, the sentence currently beginning with 'Therefore' should be the first sentence in this block of information. Followed by, the sentence currently starting with 'However' and then the one with 'Apparently'. This presents the experiments much more clearly as: i) we took care to connect things in the following way, ii) this ensures comparability, and iii) is critical in order to account for losses of strong acids to experiments surfaces.

Response: The paragraph was rearranged and literature was added as follows:

Changes: Care was taken to ensure that the connecting tubes to the entrance of the two sampling arrangements (Raschig Tube and denuder) were identical in length and diameter. This ensures that results of both sampling methods are comparable as concentration variations affect both methods equally. Although HBr concentrations between 13 and 31 ppb were established in this way, the downstream analysis revealed concentrations between 3 and 20 ppb. Probably caused by the loss of HBr on the surfaces of the experimental setup (e.g. Hanson and Ravishankara (1992), Talukdar et al. (1992), this applies generally to strong acids e.g. HNO3 (Neuman et al., 1999)). Another inaccuracy of HBr concentrations may also come from incorrect flow meters or fluctuating gas flows.

Reviewer: Page 4, Figure 1: The figure caption does not meet the journal guidelines. The panels do not get described on separate lines. Please refer to other manuscripts published in AMT to ensure correct formatting of figure and table captions. It is also difficult to see where panel (a) ends and panel (b) begins. Consider placing these in a frame or at least dividing them with a solid vertical line. Typically, the lettering for panels is placed in the upper left corner of the respective panel.

Response: Figure descriptions and panel lettering have been corrected (throughout the manuscript). Thank you for pointing this out. A dividing line between the two panels has been added in this particular figure (Fig. 1).

Changes: Modified Figure 1.

Reviewer: Page 4, Line 112: The use of 'treatment' is not very specific. Consider revising to 'denuder preparation, extraction, and analysis'.

Response: Thanks. Headline has been changed to:

Changes: 2.2 Denuder preparation, extraction and analysis

Reviewer: Page 4, Line 114: 'to avoid photochemical reactions' of what? HBr? Or photolabile HBr precursor gases, such as BrO? Or are your molecular probes susceptible to photodegradation? Please revise for clarify.

Response: This phrasing seemed to be misleading. The tubes were generally colored brown IF any coatings are used that undergo photochemical reactions. But this is not the case for the coatings described here. Therefore, the description has been deleted.

Changes: The denuder were 50 cm long brown borosilicate glass tubes with an inner diameter of 6 mm.

Reviewer: Page 4, Lines 116: Please revise for clarity. Were the '6 times of 0.5 mL' applied all at once? After each aliquot was dried? Is the drying system a custom creation or a commercial device? The use of 'therefore installed' is confusing word choice here. If it is a custom creation, a photo of the setup and a few more specific details in the Supporting Information may be of interest to others who wish to reproduce this technique or apply it to other denuder coating setups.

Response: The processing of the denuders is explained in a clearer way. The used self-made system is described in the Supplementary and shown in a photo.

Changes:

To achieve an even coating the coating solution was applied in 6 steps of 0.5 ml. Each aliquot was dried before the application of the next step. [. . .] A photo of the created drying system is shown in Fig. S3.

Supplementary:

A system for the parallel coating of four denuders has been created (Fig. S3). The system can hold four denuders at an angle of $10°$. The denuders are not fixed tight so a system of a geared motor (Modelcraft, 12 V, 2.1 A) and toothed belts can rotate the denuders with approximately 87 rounds/min. Each denuder is connected to a N2-stream of approximately 0.5 L/min. The N2-Stream is drying the applied solution and prevents leaking of the solution on the lower end of the denuder.

Alternatively, denuder can be coated handheld. The denuder has to be connected to a N2-stream. After the application of solution, the denuder has to be rotated until the solution has dried. Particular attention has to be paid on the angle of the denuder to prevent leaking.

New Figure S3.

Reviewer: Page 5, Line 123: While 'educt' technically makes sense to use here, it is fairly uncommon to encounter in this field. Consider replacing.

Response: 'educt' has been changed to 'coating reagent'

Changes: Fixation of coating reagents to the denuder walls during sampling can be done with glycerine (Finn et al., 2001) and was tested for 1,2-epoxycyclooctane-coated denuders.

Reviewer: Page 5, Lines 129-133: The use of 'elute' here is not quite correct. This term refers to the exiting of a compound from a packed column of a compound retained on a stationary phase by a mobile phase. The methodology being used here is a solvent extraction of the denuder coating and the writing should state 'extraction' instead of 'elution'.

What does 'five steps each with 2 mL of solvent' mean? Five sequential extractions, each with a volume of 2 mL for a given solvent? Was the denuder capped and inverted to ensure solvent contact with all surfaces or was the surface simply rinsed. Please clarify.

For the extraction efficiency assessment, how was the amount of 0.01 mmol EPBr applied to the denuder? It is not clear if the recovered amount from the extraction method was compared to a standard solution containing the same quantity of EPBr or if it was quantified via a calibrated instrumental technique. This should be specified.

Lastly, is the 'second elution step' being performed on a previously extracted denuder? Please clarify. Technically, this is an assessment of residual EPBr recovered in a second 5x2 mL extraction.

Response: The use of "elute" and "elution" has been exchanged for "extract" and "extraction" here and throughout the manuscript.

The extraction of the denuders and the assessment of extraction efficiency are explained in a clearer way. Fig. 2a was created for clarification.

Changes: After sampling, analytes were dissolved from the denuders in five steps using 2 mL solvent each step (Fig. 2a). All analytes were dissolved with ethyl acetate, except for EP-coated denuders, as these showed better solubility in methanol. The extraction efficiency was investigated with EP-coated denuders doped with 0.01 $\mu$mol of the bromine product EPBr. Therefore, standard calibration solution has been applied and dried on the denuder during the denuder coating process. The extraction process described in Fig. 2a has been performed a second time on a previously extracted denuder. Analyzing the residue of EPBr in the second extraction step, less than 0.05 % EPBr compared to the first step was found.

New Figure 2.

Figure caption: Figure 2: Denuder coating extraction and concentration of samples. (a)

Denuder coating is dissolved 5 times with 2 ml solvent each step. Dissolved coating is collected in glass vessels. 100 $\mu$l internal standard (NC for EP-coated denuders, TBA for all others) and 1 $\mu$l formic acid for EP-coated denuder samples is added to the approximately 10 ml coating solution before concentration. The samples are concentrated at 35 °C under a gentle nitrogen stream to approximately 100 $\mu$l. (b) The investigation of the concentration process without adding formic acid (left side) revealed a mean recovery for EPBr of 81 $\pm$ 25 % and a median (grey line) at 86%. Addition of formic acid before the concentration (right side) enhances recovery of EPBr to 99 $\pm$ 4 %. The boxes show the 25th and 75th percentiles and the median.

Reviewer: Page 5, Line 134: 'Eluates' should be 'extracts'

Response: Done.

Reviewer: Page 5, Lines 134-139: This paragraph needs reorganization for clarity. The use of formic acid appears without any motivation or reasoning, flasks are mentioned for the first time and cannot be related to prior parts of the method. The internal standards used seem to have abbreviations later in the manuscript, which are not defined here. Typically, for a concentration and recovery methodology the procedure is presented as: i) internal standard identity and quantity added, ii) conditions of the concentration step, iii) analysis. This can then be followed by the obtained recovery results without the use of formic acid, then providing the rationale for its use, and finishing with the perfect recovery results obtained.

Response: The paragraph has been completely revised for clarification. The abbreviations have been added and the beneficial use of formic acid explained. Please also refer to the newly added Fig. 2a+b (see previous comment for figure caption).

Changes: Extracts were concentrated to approximately 100 $\mu$L under a gentle nitrogen stream at 35° C. For adjustment of varying evaporated volumes and for compensation of evaporation losses, 100 $\mu$L of an internal standard was added to the extracts before the concentration step. Samples analyzed by GC-MS were doped with 100 $\mu$L 2,4,6-

tribromanisole (TBA, 6 mg/L) as internal standard, while EP-coated denuder samples analyzed by LC-MS were doped with 100 $\mu$L neocuproine (NC, 5 mg/L). The suitability of the internal standards was evaluated by investigating the recovery rate.

The recovery rate of the processing method for EPBr showed recovery rates ranging from 110 to 51%. We observed that the high amounts of EP cause precipitation when samples are concentrated for analysis. The addition of formic acid enhances solubility of analytes while not affecting negatively the following LC-ESI-MS analysis. The recovery for EPBr was determined to be 99 $\pm$ 4 % when formic acid was added to the extracts before the evaporation process (Fig. 2b).

Reviewer: Page 5, Table 1: This belongs in Section 3.1. It would be valuable to add the chemical structures of each epoxide molecular probe, and the HBr-specific reaction product, as new columns in this table. In addition, the 'result' for EP states 'suitable', but like the 9,10-epoxystearic acid a chlorine side product is mentioned in the manuscript, but not noted here. Should that not be listed here? Perhaps the column heading should be 'Observations' instead of 'Result'?

Response: Thanks for the suggestion. 'Results' has been changed to 'Observations' and moved to section 3.1. Also, the chemical structures of the coatings and their HBr-products have been added. The chlorine side-product of EP was also added. Nevertheless, EP is useful because it is less reactive compared to the 9,10-epoxstearic acid coating, but reactive enough to react with HBr.

Changes: Modified Table 1.

Reviewer: Page 6, Lines 146-147: State that this derivatization converts to carboxylate functionality to a trimethylsilyl derivative, reducing polarity and increasing volatility. Is the second sentence here supposed to be the details of the derivatization (70 C for 90 min)? Is states 'sample storage'. Suggest revising for clarity.

Response: The derivatization description has been added. Yes, 70°C and 90 min are

the derivatization details: 'sample storage' has been replaced by 'incubated'.

Changes: To increase the volatility of carboxylic acids, the carboxylate functionality was converted to a trimethylsilyl derivative by adding 30 $\mu$L N,O-bis(trimethylsilyl)trifluoroacetamide (BSTFA) and 7 $\mu$L pyridine to GC-samples containing carboxylic acids. Samples were incubated for 90 min at 70° C before analysis.

Reviewer: Page 6, Lines 151-152: Unit notation here is incorrect and inconsistent with unit presentation requirements of this journal (use a space between values and units). The separation programs in Tables S2-S3 can be combined into a single table with sub-headings for the different analytes to reduce repetition of the headers.

Response: Units have been checked throughout the manuscript. The separation programs are summarized in one table.

Changes: Modified Table S1.

Reviewer: Page 6, Lines 165-169: No sample chromatograms of the molecular probe, the reaction product, a sample collected under contrived laboratory conditions sampling HBr, nor actual samples are shown. The tables in the SI also suggest that a reference compound only noted as 'TBA' was used in these samples. What is this and why was it necessary to use? Where matrix effects were encountered, why were dilutions not performed on the samples to attempt reducing the noted effects of: shifting retention time and broad peak widths? Were the areas of the sample peaks within the linear calibrated range of the instrumentation? The described observations suggest that the column and/or detector may have been overloaded or saturated, respectively. The mass spectra for clean standard solutions of the probe and product, as well as mass spectra obtained from the field samples to confirm these molecules despite the altered retention times are necessary to include in the supporting information. One crucial chromatography metric missing from the assessment of all the methods tested is the calculated resolution between the probe molecule, the reaction produce, side products, the injection standard (presumable this is what 'TBA' is in Table S4), and the

internal standard for recovery corrections. The instrument linear range, detection limits, accuracy, and precision measured over the course of this work should all be presented in this section of the methodology for the quantified molecules noted in the preceding paragraph. It is not possible to ascertain from the current manuscript if samples were analyzed under reliable analytical conditions for quantitative analysis.

Response: The descriptions of the analytical methodology were further detailed and supplemented with figures, chromatograms and spectra. The additions can be found in sections 2.3.2, 3.2.2, S2.4, S3.1.

TBA (2,4,6-tribromanisole) and NC (neocuproine) are the two internal standards that have been used. The introduction of their abbreviations has been added in section 2.2. As explained there, it is used to compensate for variations in the volume of the concentrated samples.

Matrix effects are discussed in section 3.2.2 and supplemented with Fig. S7.

It is true that dilution of the samples would have resulted in generally lower concentrations. But it would also result in the main analyte of interest, HBr, being below the detection limit in many samples. The reviewer is also correct that the column is overloaded with the coating EP and we suspect that this is the reason for the shifted retention times. The detector, however, is not overloaded because the outlet of the LC is directed to the MS after the coating EP has passed.

Changes:

2.2 Denuder preparation, extraction and analysis

[. . .] Samples analyzed by GC-MS were doped with 100 $\mu$L 2,4,6-tribromanisole (TBA, 6 mg/L) as internal standard, while EP-coated denuder samples analyzed by LC-MS were doped with 100 $\mu$L neocuproine (NC, 5 mg/L). The suitability of the internal standards was evaluated by investigating the recovery rate. [. . .]

3.2.2 Matrix effects, precision, LOD and LOQ

The influence of abundant EP on the EPBr determination was investigated by a test series with 450 mmol/L EP (corresponds to EP concentrations in concentrated denuder samples) and EPBr concentrations in the range of 5 to 73 mg/L (n = 6, Fig. S7).

We determined a relative bias between the both sample types of $2 \pm 3$ %, concluding that no matrix effects were found.

Repeated measurements of the same gas composition using the setup shown in Fig. 1b resulted in a relative standard deviation of 8 % (see Fig. S8).

To ensure that results remain from detected analytes but not a higher noise (Fig. S5), the LOD and LOQ for denuder samples have been determined from blank denuders. The LOD and LOQ were determined by 3 and 10-fold deviation (Kromidas et al., 1995) from coated denuders transported and stored in the same way as denuder samples (3 coated but not sampled denuders for the field samples presented here). A LOD of 0.1 mg/L and a LOQ of 0.3 mg/L were calculated for EPBr. Since LOD and LOQ for HBr in the atmosphere depend on sampling time and sample volume after evaporation to concentrated samples, their values were calculated separately for each sample (Table 4).

Supplementary 2.4 Chromatograms and Spectra

Figure S5: LC-MS Chromatograms describing denuders coated with EP, for details see section 2.3.2 in the main article. (a) Extracted Ion Chromatograms of EP solution in methanol containing 7.5 mmol/L showing the m/z 197 of EP (green) at retention time 8.2 min. Samples extracted from denuders contain about 450 mmol/L. To prevent the MS from an overloading by EP, the output of the LC was led to the MS only after 17.8 min. (b) Extracted Ion Chromatograms of EPBr solution in methanol containing 1.6 mg/L showing the m/z 277 (grey) and 279 (blue) of EPBr at 27.2 min. (c) Extracted Ion Chromatograms of NC solution in methanol containing 5 mg/L showing the m/z 209 of NC (purple) at 32.4 min. (d) Selected Ion Chromatogram (using mass isolation before producing the MS) of front denuder of the field sample taken at Nindiri Rim on 18.07.16

showing EPBr (m/z 277 in grey, 279 in blue) at 25.7 min and NC (m/z 209 in purple) at 31,4 min. We assume that the shift of retention times to an earlier position is caused by the overload of the analytical column from EP-coating.

Figure S6: MS spectra, for details the section 2.3.2 of the main article. (a) MS spectrum of EP solution in methanol containing 7.5 mmol/L showing the [m+H]+ m/z 197 of EP at retention time 8.2 min. (b) MS spectrum of EPBr solution in methanol containing 1.6 mg/L showing the [M+H]+ m/z 277 and 279 of EPBr and two fragments of m/z 197 (-HBr) and 181 (-HBr-H2O) at 27.2 min. (c) MS spectrum of NC solution in methanol containing 5 mg/L showing the [M+H]+ m/z 209 of NC at 32.4 min. 3.1 Matrix effects and precision

Figure S7: Analysis of matrix effects comparing Samples containing EPBr concentrations with 450 mmol/L EP (+EP) and without EP (-EP) in the sample. The orthogonal distance regression model results in (+EP) = 1.03 x (-EP) − 0.65. The relative bias between the both sample types is $2 \pm 3$ %.

Reviewer: Page 7, Line 173: 'a significant loss of the compound was observed during sampling' How? Was this quantified by an instrumental method? What was the exact amount lost? Everything? How was 'significance' determined statistically?

Response: There was nothing left of the coating material 1,2-epoxycyclooctane. Traces of the bromohydrin were found but we did not quantify the product formation.

The statement has been clarified:

Changes: Although it is solid at room temperature, 1,2-epoxycyclooctane was not detectable in samples extracted from denuders after sampling. Nevertheless, traces of the product 2-bromocyclooctanol could be detected in the denuder extracts. We did not quantify the formation, but this led us to believe in the principle ability of epoxides to react with HBr in our system.

Reviewer: Page 7, Line 174: 'Obviously, 2,3-.... Is too volatile.' is redundant. Delete.

Response: Done.

Reviewer: Page 7, Lines 174-177: Provide quantitative information to support your conclusion. There are too many ideas in a single sentence here and the authors should attempt to separate these for clarity.

Response: Please see previous comments.

Reviewer: Page 7, Line 178-179: 'massively' and 'however' are not needed. Delete.

Response: Done.

Reviewer: Page 7, Line 183: Should 'causes low volatility' be 'reduces volatility'?

Response: Done

Reviewer: Page 7, Lines 187-188: 'Although derivatized HBr...'. Table S4 states that these molecules had different retention times, but resolutions between the peaks are not given. What is the actual analytical issue here? And if the analytes could be quantified, what does it matter if the products all have the same m/z ion in EI if you can identify the product by injecting an authentic standard? Presumably, if there are matrix effects a spike of the product through standard addition would be a viable option, but would be more labour intensive. This discussion should be expanded and be more thorough.

Response: We added sample chromatograms to the supplementary material for clarification (Fig. S.4). As a coating, this substance also seemed to react readily with water, which occurs in excess in volcanic plumes compared to HBr. A standard addition would certainly be necessary here for reliable quantification. While these samples have to be additionally derivatized because of the carboxylic acid and the concentrated sample volumes are generally very small, performing a standard addition appeared impractical in terms of errors in the workup and possible contamination.

Therefore, we preferred EP, which also reacted successfully with HBr under the given

conditions.

Changes: Although derivatized HBr could be analyzed, the m/z ratios were overlaid by the water and chloride derivatives (same main m/z ratios, Fig. S4). The difficulties this posed for analysis led us to prefer EP, as it also reacted successfully with HBr in our system.

New Figure S4.

Figure caption: Figure S4: GC-MS Chromatograms following temperature program B describing denuders coated with 9,10-epoxystearic acid. (a) GC-MS chromatogram in full scan showing retention times for oleic acid (5.99 min), 9,10-epoxystearic acid (6.87 min), 9,10-dihydroxystearic acid (7.43 min), and 10-bromo-9-hydroxystearic acid (8.13 min). (b) GC-MS chromatogram in full scan showing retention timed for oleic acid (6.00 min), 9,10-dihydroxystearic acid (7.44 min), and 10-chloro-9-hydroxystearic acid (7.79 min). (c) GC-MS chromatogram with selected ion monitoring (SIM) of m/z 317 of a sample collected with 9,10-epoxystearic acid coated denuder on Mt. Etna in 2015 showing broad peaks between minute 7 and 8 presumably caused by 9,10-dihydroxystearic acid and 10-chloro-9-hydroxystearic acid. Note that the coating 9,10-epoxystearic acid disappeared, presumably has been used up by the reaction with water and HCl. (d) zoom of chromatogram (c) that shows 10-bromo-9-hydroxystearic acid the product of the coating with HBr and the high and variable background. We assume that the shift of retention times to an earlier position is caused by the overload of the analytical column from the coating.

Reviewer: Page 7, Lines 189-190: Certainly more discussion and the criteria for meeting the designation 'suitable' are warranted here!

Response: The details why we consider this method 'suitable' are of course part of the next sections. We have now included a connection set as a transition.

Changes: EP as a coating agent could retain and derivatize gaseous HBr passed

through denuders and the bromohydrin product could be detected. The details of the characterization for this coating-compound are given in the following sections.

Reviewer: Page 7, Line 190: 'chapters' should be 'sections'.

Response: Done.

Reviewer: Page 7, Section 3.2: There are many different topics within this section that would greatly benefit from separate subheadings (e.g. Coating Breakthrough Experiments, QA/QC and Matrix Effects, Coating Stability, and Interferences and Cross Reactions). It would help organize the section and reduce the number of places where connecting sentences would otherwise be required to make the discussion contents easier to follow.

Response: We agree with the reviewer that subheadings makes the manuscript easier to follow and the following headings were added: 3.2.1 Collection efficiency, 3.2.2 Matrix effects, precision, LOD and LOQ, 3.2.3 Stability of extracted samples, 3.2.4 Interferences

Reviewer: Page 7, Line 196: 'Therefore, the' can be 'We tested'. The rest of this sentence requires further clarification. One can eventually discern that the point is to apply an upper limit of HBr mixing ratio to the denuder, based on prior ambient observations of plumes.Suggest revision for clarity.

Response: The whole section was revised. It is true that the HBr concentrations used are far above those expected in the field to test possible maximum values. Since the same procedure was used, the collection efficiency alongside HCl was moved here. Table 2 has been added to make the whole procedure more understandable.

Changes: According to Rüdiger et al. (2021) and Wittmer et al. (2014), about 0.5-5.9 ppb and 9.5-36 ppb total bromine was detected in ground-based samples in the volcanic plume of Masaya and Etna volcano respectively using alkaline traps. Besides HBr they detected HCl concentrations of 0.5-4.5 ppm and 0.1-20.6 ppm respectively.

HCl can react with EP with the same reaction pathways as HBr and thus consume the coating reagent. Other halogen species such as HCl form different derivatization products with EP and can, therefore, be easily distinguished by mass spectrometry. Estimating the speed of the derivatization reaction the nucleophilic reactivity of different hydrogen halides shows that bromide has higher nucleophilic reactivity than chloride (nucleophilic constants in water based on glycidol, H2O: 0.00, Cl-: 3.04, Br-: 3.89, I-: 5.04) (Swain and Scott, 1953). Accordingly, one could expect a higher reactivity of HBr compared to HCl.

New Table 2.

Reviewer: Page 7, Line 199: How was the coating quantity on the denuders determined? Was this quantified using the instrumental methods presented or was this determined by calculation? From the description of the denuder coating section, it does not seem likely that all of the analyte solution applied to the denuder surface is retained and that likely a large amount of the applied solution exits the denuder when it is rotating on the 10 degree sloped setup with gas flowing over it. This contrast between theoretical amount of probe applied compared to the practical result of the method would be valuable to discuss here if the applied amount was not quantified.

Response: The exact quantity of coating amount on the denuders has not been determined. The reviewer is right, the exact amount may vary between denuders. An excessive loss of coating reagents would appear as contamination in the denuder mounting system during denuder preparation and has not been observed.

Because the coating is present in abundant amounts we believe that the variations in total amount have negligible effects on the analyte retainment. Rather we suspect that the better retainment of analytes with higher coating amount is mainly caused by a better/more homogeneous covering of denuder walls.

To overcome any misleading phrasing, we changed phrases describing exact amounts on denuders to name the used coating solution.

Changes: e,g.

The collection efficiency tested for denuders coated with 7.5 mmol/L EP coating solution revealed a breakthrough of HBr since about 30 % of the amount of the first denuder was observed in the third denuder.

Reviewer: Page 8, Line 200: 'clearly' is not required. Delete.

Response: Done.

Reviewer: Page 8, Line 201: 'was below 1 %' in which denuder? The third one? For all field samples collected were both denuders extracted to ensure accurate quantitative collection of the targeted HBr? The following use of 'however' can be deleted.

Response: Table 2 has been included to make the procedure comprehensible (Please see the previous comments).

Yes, for all field samples both denuders were extracted and summarized. This description has been added to field sampling description (section 2.4). Most of them were below their limit of detection anyway.

Changes: In section 2.4

. . . Both denuders were extracted and results were summarized. . . .

Reviewer: Page 8, Line 202: Given the issues encountered with peak width and shape in the HPLC separation, perhaps such aggressive concentration steps were not required to retain suitable detection limits for the method? It would be worthwhile to comment on such findings here. The use of 'finally' at the end of this sentence does not really work. Consider using 'We determine that use of 15 mmol/L EP coating solution was ideal to coat denuders with 45 umol EP.'

Response: The reviewer is right. The high amount of EP as a coating is a double-edged sword. The high amounts of EP can cause preciptitaions when samples are concentrated for analysis but are needed to ensure complete retaining of all HBr during

sampling. We further need that excessive concentration-step to make sure that the Br-derivative EPBr (the analyte of interest) can pass the LOD/LOQ. The more the samples are concentrated (the more concentratred is EPBr) the more likely it is to exceed the LOD/LOQ of EPBr.

Changes: Concluding from this, 15 mmol/L EP coating solutions were used to coat denuders with a theoretical amount of 45.0 $\mu$mol EP.

Reviewer: Page 8, Lines 204-206: 'The influence of EP' due to what? Its much higher concentration relative to EPBr? What are the 450 mmol/L 'concentrations'? Are these standard solutions? Were the mixtures done at equal amounts of EP compared to EPBr? A retention effect of EP on EPBr is likely to be muted under equimolar mass loadings on the analytical column and observed more realistically by simulating ratios you'd expect to find in real samples. What does the 'n=6' refer to here? Six standard mixtures of equimolar concentration between the upper and lower bounds stated? Six injections of the two concentrations? Does the upper limit of this range correspond to the maximum concentration of EP that could be extracted from a denuder? None of these important details are discussed, but they have surely been considered. Please expand the discussion to improve clarity and communicate the care and detail considered in this work.

What values were the standard deviation percentages determined from? Some metrics are very useful (e.g. accuracy and precision in quantitation) while others are not (e.g.retention time). This is critical to specify and justify why the metric you selected has utility in concluding that no matrix effects exist in the separation technique from the probe and product molecules.

Response: The ranges tested for the influence of EP on EPBr refer to the concentration of EPBr, EP was applied in the concentration of 450 mmol/L which imitates the concentrated extracts of a denuder coated with 15 mmol/L EP coating solution (described in new Table 2).

The test sample with the lowest concentration of 15 nmol/mL corresponds to a real sample of 1.5 nmol/extracted sample. It is true that this only include higher concentrated field samples (see Table of field samples). We believe that a strong biasing effect should be visible even at high concentrations.

'n = 6' has described the number of compared samples. More clearly to see now in Fig. S7.

The phrasing 'deviation' was misleading here, we calculated the difference between the results of both methods. The statement has been revised.

We suspect that the shifts in retention time result from an overloading of the column from excessive EP amounts. But here we show, that despite a higher noise (which causes a higher LOD/LOQ) we can quantify EPBr without issues.

Changes: The influence of abundant EP on the EPBr determination was investigated by a test series with 450 mmol/L EP (corresponds to EP concentrations in concentrated denuder samples) and EPBr concentrations in the range of 15 to 265 nmol/mL (n = 6, Fig. S7).

We determined a relative bias between the both sample types of $2 \pm 3$ %, concluding that no matrix effects were found.

New Figure S7.

Reviewer: Page 8, Line 207: 'Constricted' should be 'concentrated'.

What are the 'samples' mentioned here? Contrived samples made from pure compounds, field samples, lab samples, or all of the above? How many times was the sample reanalysis performed and at what temporal frequency (e.g. monthly, weekly)? A plot of the data and clear indication of the identity of the samples is required to ascertain whether the stated stability of the probe is a sound conclusion. The trend of sample stability/degradation quantified should be presented with a figure in the Supporting Information.

Response: Only the short-term change in the samples was systematically studied. We only observed the long-term effect on the basis of individual measurements of the field samples. We have created a table and a figure to clarify the samples and observations (Table S3 and Fig. S9).

Changes: We tested the stability of extracted and concentrated samples with field sample like approaches that were stored in the freezer at $4°$ C (Table S3 and Fig.S9). Within the first two months of storage no systematic loss could be observed when comparing the analyzed EPBr/NC ratios. After long term storage of 2-3 years remeasuring the field samples (listed in section 3.5) revealed an average loss of $0.03 \pm 0.01$ % EPBr per day (about 11 % loss after 1 year of storage).

Figure S9: Stability of extracted and concentrated samples. The samples were stored in the freezer at $-4°$C. Stability of field-like lab samples during the first 80 days. Assignment of the samples can be seen in Table S3. The regression model indicates that a collective significant effect was not found. The field samples described in section 3.5 have been measured again after 2-3 years. All samples revealed a loss of -20 to -40 %. That refers to $0.03 \pm 0.01$ % Note that the x-axis has a logarithmic scale.

Reviewer: Page 8, Lines 211-215: The authors present method detection and quantitation limits, but instrument performance metrics are missing from the methods section (as well as how they were determined). Since the molecular probes are novel analytes without pre-existing methods published in the literature, such QA/QC is critical to present here. Please add, so that the results presented here can be evaluated with that information in mind. Were there detectable peaks for the reaction product in the field blanks that set the method detection limits higher than the instrument detection limits? How many field blanks were analyzed to calculate these values? What was the variance between the blanks, if any?

Were there detectable reaction product peaks in a denuder that was extracted immediately after drying? How did these compare and what does this tell you about the

potential for contamination of denuders during transport? This helps clearly set up the reasoning behind the assessment of interferences and cross-reactivity that follows.

Response: There were no obvious peaks in blank samples, but increased noise across the entire chromatogram in all denuder samples, presumably from column overloading. 'All denuder samples' means samples concentrated after extraction from EP-coated denuders, i.e. field samples, laboratory samples and coated blank-denuders without any sampling. To make this visible to the reader, an example chromatogram is shown in the supplement (Fig. S5d). Therefore, we did not use the LOD that resulted from the calibration line, but the reference to the standard deviation of the blanks suggested by Kromidas et al. 1995. The LOD and LOQ of the calibration line were nevertheless added in the HPLC part.

For the field samples shown here, there were 3 blank denuders that accompanied the field samples coated but unsampled. This information has been added to section 3.2.2.

The variation between the blanks is reflected in the LOD and LOQ, both calculated from the standard deviation of the blanks. The standard deviation was 0.03 mg/L. For some experiments we found high values Cl- and Br- derivates on blank denuders (experiments not shown here). Especially on denuders that have been stored unprotected (only covered with caps) in the laboratories (Obviously there were HCl and HBr in the lab air reaching the denuder coating). Learning from these observations denuders were sealed carefully and stored in the freezer. Also, all experiments/field samples were accompanied by blank denuders all time to -at least- notice any contamination. For the field samples shown here no Br-contamination was observed.

Changes:

Section 3.2.2

[...] To ensure that results remain from detected analytes but not a higher noise (Fig. S5d), the LOD and LOQ for denuder samples have been determined from blank denuders. The LOD and LOQ were determined by 3 and 10-fold deviation (Kromidas et al., 1995) from coated denuders transported and stored in the same way as denuder samples (3 coated but not sampled denuders for the field samples presented here). A LOD of 0.1 mg/L and a LOQ of 0.3 mg/L were calculated for EPBr. Since LOD and LOQ for HBr in the atmosphere depend on sampling time and sample volume after evaporation to concentrated samples, their values were calculated separately for each sample (Table 4).

Reviewer: Page 8, Line 217: 'would be at least conceivable as a' is very confusing. Consider 'isthe most likely' instead. 'However' at the end of this line is, again, unnecessary.

Response: Done.

Reviewer: Page 8, Lines 217-218: What mixing rations of Br2 were used, what other gases were present with the Br2? This sentence requires revision for clarity and justification for the representativeness of the Br2 quantities investigated.

Response: The information about the output rate of the Br2 source has been added to the description of test gas sources. The denuders were connected to the source without further dilution as shown in Fig. 1a. This means that the concentrations administered are far above those expected in field samples. No bromine product was observed in any of the denuders.

Changes:

2.1.1 Test gas sources

[...] For Br2, the permeation source described in Rüdiger et al. (2017) was used with an output rate of 775 $\mu$g/d under nitrogen flow.

[...]

3.2.4 Interferences

If the denuder coating reacts with other types of bromine, resulting in the same product, this would lead to HBr overdetermination. Elemental bromine (Br2) is the most likely cross-interference. 9,10-epoxystearic acid and EP-coated denuder collected the output of the Br2 source for one hour following the setup in Fig. 1a. No bromine product was found. Other bromine species such as bromine oxides (e.g. BrO) with their positively polarized bromine atoms, no nucleophilic attack on the epoxide reaction center leading to bromohydrin is expected.

Reviewer: Page 8, Lines 219-221: Since methyl bromide was not explicitly tested, has not beendetected in volcano plumes, and will not produce the same product as HBr, it is not worth discussing. Delete.

Response: Done.

Reviewer: Page 8, Line 228-230: Reorganize and clarify.

Response: The whole section has been rearranged and rewritten for clarification. Changes: According to Rüdiger et al. (2021) and Wittmer et al. (2014), about 0.5-5.9 ppb and 9.5-36 ppb total bromine was detected in ground-based samples in the volcanic plume of Masaya and Etna volcano respectively using alkaline traps. Besides HBr they detected HCl concentrations of 0.5-4.5 ppm and 0.1-20.6 ppm respectively. Other halogen species such as HCl react with EP via the same reaction pathways as HBr. But these form different derivatization products with EP and can, therefore, be easily distinguished by mass spectrometry. Still, HCl can consume the coating reagent. Estimating the speed of the derivatization reaction the nucleophilic reactivity of different hydrogen halides shows that bromide has higher nucleophilic reactivity than chloride (nucleophilic constants in water based on glycidol, H2O: 0.00, Cl-: 3.04, Br-: 3.89, I-: 5.04) (Swain and Scott, 1953). Accordingly, one could expect a higher reactivity of HBr compared to HCl.

New Table 2.

Reviewer: Page 8, Lines 230-231: How long as the 4 ppm HCl exposure performed for? A comparable duration to field sampling volume and duration? What about integrated exposure with some approximated limits based on the literature?

Response: This part of the section has also been rewritten and reordered. Table 2 has been added to make the experiments easier to follow. Some of the literature on expected HCl concentrations was already given in the manuscript, but elsewhere. All thoughts on HCl have now been summarized in this section (see details also in previous comment).

Changes: Ensuring that the method will be able to retain all potential gaseous HBr even in high concentrated plumes the breakthrough behavior for 0.2 $\mu$mol HBr was investigated (0.2 $\mu$mol HBr correspond to 1 ppm HBr for 1 h sampling duration). Further, the collection efficiency for HBr was tested in the presence of about 5 $\mu$mol HCl (5 $\mu$mol HCl correspond to 4 ppm HCl for 1 h sampling duration).

The collection efficiency was tested with two or three denuders connected in series (Table 2). The amount of product found on the second and third denuder was compared with the values of the first denuder. The collection efficiency tested for denuders coated with 7.5 mmol/L EP coating solution revealed a breakthrough of HBr since about 30 % of the amount of the first denuder was observed in the third denuder. In contrast, for denuders coated with 15 mmol/L EP coating solution, the breakthrough for 1 ppm HBr was below 1 %. In competition with HCl 1.9 $\pm$ 0.4 % of the bromine product was found in the second denuders.

Coating amounts above 45 $\mu$mol EP caused precipitation in concentrated samples during sample preparation (see section 2.2). Concluding from this, 15 mmol/L EP coating solutions were used to coat denuders a theoretical amount of 45.0 $\mu$mol EP. The second denuder in series during sampling ensures that we will at least notice a relevant breakthrough of analytes.

New Table 2.

Reviewer: Page 9, Lines 238-240: The alkaline traps are the established techniques and the new EP probe should be compared against them. Here, the comparison is made the other way around.

Response: Done.

Changes: On average, the results related to denuder sampling yielded 99 $\pm$ 11 % of the HBr values determined by the Raschig Tube.

Reviewer: Page 9, Figure 3: This should be converted to a table with the +/- values given and statistical tests for similarity performed to demonstrate that both methods are measuring the same quantity of HBr. This is a more robust validation of the developed EP molecular probe. The last sentence in the caption is a repetition of text already presented in the discussion. Delete.

Response: This comment highlighted a clear gap in our discussion. A table with the determined values was added for a better overview. An orthogonal distance regression was performed for comparing the results of both methods.

Changes: Alkaline traps determine the total bromine content, gaseous HBr is thus measured here as a part of the total bromine. Speciation of the individual bromine species is not possible with alkaline traps, but if only gaseous HBr is sampled by both methods in the laboratory, denuder method and alkaline traps should produce the same results. Therefore, a comparative measurement of the denuder method with a Raschig Tube as an alkaline trap was set up to check the accuracy of the newly developed denuder method. The experimental setup is shown in Fig. 1b. In five experimental series, HBr concentrations between 3 and 20 ppb were determined simultaneously (Table 3). A Dean-Dixon outlier test was applied in order to evaluate possible outliers (Dixon, 1950). No outlier was identified for $\alpha$=0.05 (If the significance level was changed to $\alpha$=0.01, experiment 4 could be considered an outlier.). Consequently, all the results were taken into account.

On average, the results related to denuder sampling yielded 99 $\pm$ 11 % of the HBr values determined by the Raschig Tube. An orthogonal distance regression was performed and is shown in Fig. 4. Based on the line equation obtained, small values, such as those observed here in the field samples, can yield higher results from denuder determinations than expected via the Raschig Tube. We have concluded, that the HBr values determined by denuders in field samples can be considered a fraction of the total bromine determined by the Raschig Tubes. To account for the comparison studied, the deviation found is included as an error of the denuder field samples in Table 4.

New Table 3.

Modified Figure 4.

Reviewer: Page 9, Section 3.5: Much of the information presented here should be relocated tothe methods section to describe the sampling site and sampling approach. Only results and discussion of the new denuder method should be presented here.

Response: The description of the field work has been moved to the new section '2.4 Field application at Masaya 2016'.

Changes:

2.4 Field application at Masaya 2016

A first set of field samples was collected between 18.-21. of July 2016 at the Santiago Crater of the Masaya volcano (Nicaragua). A detailed description of the location can be found in Rüdiger et al. (2021). In summary, sets from different methods were collected simultaneously together at changing locations with various distances (200-2000 m, Fig. 3b) to Masaya's emission source at Santiago crater (Fig. 3b).

A total of eight ground-based and two UAV-based samples for the newly developed denuder method are presented here. In ground-based sampling sets, two denuders were sampled in series (Fig. 3a). Both denuders were extracted and results were summarized. Sampling was performed by a Gilian GilAir Plus handheld pump (battery

included) with a flow rate of 250 ml/min for about 1-1.5 hours for each denuder. In addition to EP-coated denuders, samples with 1,3,5-trimethoxybenzene coated denuders for the determination of reactive bromine. as well as Raschig Tubes as alkaline traps for the determination of total bromine and total sulfur were collected simultaneously side by side. The results of these samples can also be found in Rüdiger et al. (2021).

First drone-based samples were collected with an UAV using a small four-rotor multi-copter with foldable arms (Black Snapper, Globe Flight, Germany) called RAVEN (Rüdiger et al., 2018). For the UAV-based sampling, a remotely controlled sampler (called Black Box) was used and is also described in detail in Rüdiger et al. (2018). The Black Box enabled logging of the sampling duration and SO2 mixing ratios via the built-in SO2 electrochemical sensor (CiTiceL 3MST/F, City Technology, Portsmouth, United Kingdom). The Black Box has 20x14x13 cm. With this setup (Black Box + denuder) of approx. 1 kg we achieved flight times of up to 15 min. In drone-based sampling flights, individual denuders were used with sampling times between 5-10 minutes.

Modified Figure 3.

Figure caption: Figure 3: Field campaign at Masaya volcano in July 2016. (a) Sampling setup containing TMB- and EP-denuders, two in series each, and Raschig Tube. (b) Overview on ground-based sampling locations (purple areas) during the field campaign at Santiago crater, on Nindiri rim and UAV-based (blue area) in the caldera valley.

Reviewer: Page 9, Lines 251 and 256-257: The UAV system needs to be briefly summarized here. Referring to prior reports alone is not adequate.

Response: Done. Please see previous comment.

Reviewer: Page 9, Line 255: The NOVAC station is not defined anywhere? What does this standfor? Figure 4 does not denote it specifically. In the subsequent discussion, there is nocomparison made to the DOAS measurements noted here. This is a highly valuablefield intercomparison. What were the results or why was it not possible to get

usabledata? Ideally, this should be expanded.

Response: Since the BrO data obtained by DOAS are already processed in the model run used here for comparison and the data do not otherwise provide any further means of comparison, they have been removed from this manuscript.

Reviewer: Page 10, Figure 4: Color code the ground versus UAV locations. Place distances fromthe HBr emission source to the sampling locations on the map and remove from Table2. Response: Done. Please see the previous comments.

Reviewer: Page 10, Line 262: The range is from '< LOD' to 1.97 ppb. A reference to Table 2 should be made at the end of this sentence and the last sentence of this paragraph deleted.

Response: The ground-based samples mentioned in this sentence were all above the LOD. Only UAV-based samples (next sentence) were below their LOD.

Changes: Here we present a first set of field samples using the new denuder method. The eight sets of ground-based measurements range from 0.44 to 1.97 ppb (Table 4). Two UAV-based measurements are below their LOD. In UAV-based samples, the higher LOD in UAV-based measurements result from a much shorter sampling time of 5-10 minutes (limited by maximum possible flight time) compared to ground-based measurements (1-1.5 h).

Reviewer: Page 10, Lines 264-265: This information belongs in the caption for Table 2. Relocate.

Response: Done.

Reviewer: Page 10, Line 268: The 2x2 information in brackets should be deleted and a column for the number of samples collected at each location added to Table 2.

Response: The two pairs of samples were entered more clearly in the table. Please see comment below.

Reviewer: Page 10, Line 268-270: are these mixing ratios statistically the same or different? Can the precision of the method be approximated despite the small number of replicate samples? This is an example of where determination of the method precision for the laboratory experiments can bolster the robustness of the method when applied in the field and the discussion here can bridge those findings together.

Response: We compared the relative standard deviation of the parallel data sets with that of laboratory samples that also sampled the same gas composition.

Changes: To get an idea about the reproducibility of these field measurements a duplicate set of two EP-coated denuders each was collected in parallel side by side on July 21. The parallel denuder measurements resulted in HBr concentrations of 1.97 ± 0.11 and 1.82 ± 0.10 ppb at the Santiago rim and 1.17 ± 0.07 and 0.97 ± 0.09 ppb at the Nindiri rim. The mean values and standard deviations of 1.90 ± 0.11 and 1.07 ± 0.14 ppb of the two parallel samples result in a relative standard deviation of 6 and 13 %, respectively. While the deviation from the Santiago rim samples is within those also found for laboratory samples (8 %), possible causes leading to larger errors and affecting simultaneously collected samples differently may be passive diffusion during installation of the tubes or ash blowing in.

Reviewer: Page 10, Line 270: A deviation of a deviation is not very meaningful and a good reason for calculating it is not given. Justify or remove.

Response: Deleted.

Reviewer: Page 10, Line 276-282: The model-measurement comparison is too brief and needs to be expanded to clearly demonstrate the quality of the field measurements. For example, the loss of HBr in the model is stated as a percentage, but the loss of HBr observed is discussed in the same context. One has to hope they identify the correct information in Table 2 and then calculate this independently to ascertain the statemen of 'very good agreement' is true. Overall, I cannot follow the logic of this section and it weakens the manuscript when it should be strengthening it. Please

revise.

Response: Apart from rewriting the paragraph, the total bromine and HBr/total Br ratios have been added to the table to make it easier to follow. In addition, the model run mentioned was taken over into the figure.

Changes: In the work of Rüdiger et al. (2021) the results of total bromine, total reactive bromine and bromine monoxide from accompanying methods were used to run the atmospheric box model CAABA/MECCA, which was initialized by a high-temperature equilibrium model. The model run that best described the data in Rüdiger et al. (2021) was used here for comparison and is highlighted in light blue for the ratio of HBr/total Br in Fig. 5. This run was based on a Br/S ratio of 7.4 x10 4. The Br/S ratio for the measurements considered here on 18.-21.7. was on averaged 6.2 $\pm$ 1.0 x10 4. Even though the general trend between measured values and model predictions is consistent, on average the measured values appear to be slightly higher than those calculated by this model run. Following the observations in Rüdiger et al. (2021), a cause may be the influence of aerosol. Aerosol was not measured simultaneously, smaller particle number concentrations and diameters than assumed may lead to slower HBr loss than expected. Also, deviation from the assumed wind speed can lead to a horizontal shift of the measurements while the deviations between Denuder and Raschig method cause a vertical shift. Overall, the trend observed would have to be confirmed by further samples. These samples give us a first idea that we can confirm our general idea about the HBr consumption. Of course, a solid foundation will require many field samples and further consideration of the two methods used and their joint application at the expected concentrations.

Modified Figure 5.

Figure caption: Figure 5: Fraction of HBr (determined by EP-coated denuders) to total bromine (determined by Raschig Tubes). Assuming a windspeed of 5 m/s, HBr fractions decrease on average from 0.75 $\pm$ 0.11 at 0.7 min at Santiago Rim to 0.36 $\pm$

0.08 at 2.5 min on Nindiri Rim. The colored area in light blue describes the fraction of HBr calculated by the model of Rüdiger et al. (2021) (model parameter to identify the selected run: An initial ratio of 10:90 of atmospheric and magmatic gas was assumed at high temperatures. The output was then quenched to 30 ppm SO2 for the start of the low temperature chemistry. The proportions of hydroxyl radicals (OH), hydrogen peroxide (H2O2) and hydroperoxyl radicals (HO2) and nitric oxide radicals (NOx) correspond to the atmospheric background composition. Within 10 minutes, dilution by a factor of 1/e (0.37) occurred. The number of particles per m3 was 3 x109, their radius 300 nm.)

Reviewer: Page 11, Table 2: Samples with HBr quantities below the method detection limits should be reported as '<LOD' or as '<XX' for the corresponding detection limit of a given sample. A column for the number of samples at each location and date needs to be added, as well as for the sampling method (ground vs UAV). Is the Br/S in the table measured or modeled? This needs to be stated in the figure caption and in the discussion.

Response: Additions have been made to the table. Values below the LOD have been indicated, UAV-based and simultaneously measured samples have been marked.

As supporting data, the results of the Raschig Tubes for the respective samples were used (total bromine and also Br/S). The values of the Raschig Tubes have already been published in Rüdiger et al. (2021). This was already noted with an '*' in the table, but has now been additionally added to the table heading for clarification. It is also mentioned in the text where details of the Raschig Tube results can be found.

Changes: Modified Table 4.

Reviewer: Page 11, Figure 5: Can the fraction of HBr/Br from the model be added to the figure for comparison? Based on the discussion this should be possible and compare well.

Response: Yes. We liked the idea to add the model to the figure.

Changes: Modified Figure 5.

Conclusions: Rewrite based on revised manuscript and in light of all comments above.

Please also note the supplement to this comment:
https://amt.copernicus.org/preprints/amt-2020-357/amt-2020-357-AC2-
supplement.pdf

[Figure]

**Fig. 1.** Experimental sampling setups in the laboratory. A separating line has been introduced between both setups.

[Figure]

**Fig. 2.** New figure to clarify processing. For full caption please see text.

[Figure]

[Figure]

**Fig. 3.** Ground-based and UAV-based sampling areas have been colored differently. For full caption please see text.

[Figure]

**Fig. 4.** Denuder results are plotted against Raschig results. The orthogonal distance regression model resulted in: y = 0.97 (± 0.10) * x + 0.18 (± 0.10) and a residual variance of 0.25.

[Figure]

**Fig. 5.** The model run has been added to the figure. For full caption please see text.

[Figure]

**Fig. 6.** = Figure S2: Output of diffusion gas sources of 48% HBr at 30 °C (black), 48% HBr at 50 °C (yellow) and 30% HCl at 50 °C (blue).

[Figure]

**Fig. 7.** = Figure S3: Created system (called "Denudermaster") for the simultaneous coating of four denuders.

[Figure]

**Fig. 8.** = Figure S4: Chromatograms describing describing denuders coated with 9,10-epoxystearic acid. For full figure caption please see text.

[Figure]

**Fig. 9.** = Figure S5: Figure S5: LC-MS Chromatograms describing denuders coated with EP. For full figure caption please see text.

[Figure]

**Fig. 10.** Figure S6: MS spectra describing denuders coated with EP. For full figure caption please see text.

[Figure]

**Fig. 11.** Figure S7: Analysis of matrix effects. For full figure caption please see text.

[Figure]

**Fig. 12.** = Figure S8: Precision of the method was determined by analysing five times the same test gas denuders. For full figure caption please see text.

[Figure]

**Fig. 13.** = Figure S9: Stability of extracted and concentrated samples. For full figure caption please see text.

**Supplement:**

**Modified tables in response to Review 1**

**Table 1: Selected epoxides used as coating reagentstable**

[revised manuscript text omitted]

 **Table S 1: GC-temperature programs**

| Heating rate [°C min$^{-1}$] | End temperature [°C] | Holding [min] | Duration [min] |
|---|---|---|---|
| Program A, analysis of 1,2-epoxycyclooctane coated denuders | | | |
| | 90 | 3.00 | 3.00 |
| 38 | 210 | 0.00 | 6.16 |
| 9.5 | 235 | 0.00 | 8.79 |
| 30 | 250 | min. 2 | min. 11.29 |
| Program B, analysis of 9,10-epoxystearic acid-coated denuders | | | |
| | 120 | 0.50 | 0.50 |
| 38 | 250 | 0.00 | 3.92 |
| 10 | 300 | 0.00 | 8.92 |
| program C, analysis of trans-oxirane-2,3-dicarboxylic acid and 3-Phenyloxirane-2-carboxylic acid coated denuders. | | | |
| | 90 | 3.00 | 3.00 |
| 18 | 150 | 3.00 | 9.33 |
| 25 | 250 | 17.00 | 30.33 |

---

## Author Response (AR2)

**Author's response letter – amt-2020-357**
**"Bromine Speciation in Volcanic Plumes: New in-situ Derivatization LC-MS Method for the Determination of Gaseous Hydrogen Bromide by Gas Diffusion Denuder Sampling"**

5 # by Gutmann et al.

**Report #1 - Response to questions and comments from Reviewer #4**

The manuscript has been greatly improved. Only some minor comments should be addressed before publication.

We thank the reviewer for the time to review this manuscript and the suggestions for improvement.

1. Line 24 "HBr concentrations in the range between 0.44 and 2.27 ppb were measured", but as shown in Table 4, this value

10 ranges from 0.44 to 1.97 ppb.

The value in the abstract has been corrected.

**The denuder sampling setup was applied in the plume of Masaya (Nicaragua) in 2016. HBr concentrations in the range between 0.44 and 1.97 ppb were measured with limits of detection and quantification below 0.1 and 0.3 ppb respectively.**

15 2. Figure S6 is not mentioned in the manuscript.

A reference to S6 has been added to the manuscript.

**Under these conditions, EP eluted at 7.9 min retention time, however, with relatively broad peak widths due to the high concentration of the coating material in the concentrated samples. 5-chloro-6-hydroxy-5,6-dihydro-[1,10]-phenanthroline eluted at retention time 20.5 min. In the field samples, the peak widths increased and retention times**

20 **changed, probably due to overloading of the column. EPBr eluted at retention time 27.2 min. Depending on the amount of coating material and chloro-derivative, the retention time varied between 25 and 28 min. The internal standard neocuproine eluted at retention time 32.4 min (Fig. S5 and S6).**

3. Line 273 Keep the same unit for EPBr with Figure S7.

The units were converted accordingly.

25 **The influence of abundant EP on the EPBr determination was investigated by a test series with 450 mmol/L EP (corresponds to EP concentrations in concentrated denuder samples) and EPBr concentrations in the range of 19 to 263 nmol/ml (n = 6, Fig. S7).**

**Report #2 - Response to questions and comments from Reviewer #3**

30 The Authors have made substantial changes and these have dramatically raised the quality of this work! Such comprehensive responses and high quality additions to a manuscript certainly required a lot of effort. The extensiveness of the response to the Reviewers is an uncommon demonstration of dedication by these scientists to highly rigorous work. The resulting manuscript is well within the scope of Atmospheric Measurement Techniques and is suitable for publication. Thank you for the pleasure of this!

35 We thank the reviewer for the interest on our manuscript and for recognizing our efforts.

One extremely small issue remains in the revised manuscript, that would be worth clarifying is:

The authors state that matrix effects do not exist and that is not technically true. There is a matrix effect in the method that impacts the retention of the analytes, but not the ability of the method to quantify the reaction product. This is because the

40 mass spectrometer detects their unique mass fragments and when coupled with the internal standard it makes taking the matrix effects into account possible with very little error. If the authors could rephrase this, it would be more accurate.

That is true. We have improved the wording.

**We determined a relative deviation between the two sample types of 2%. We concluded that due to the detection of individual mass fragments by the mass spectrometer and internal standard adjustment, the ability of the method to**

45 **quantify the reaction product is maintained even with EP-matrix.**